



# Modern silicon dynamics of a small high-latitude subarctic lake

Petra Zahajská[1,5], Carolina Olid[2], Johanna Stadmark[1], Sherilyn C. Fritz[3], Sophie Opfergelt[4], and Daniel J. Conley[1]

[1]Department of Geology, Lund University, Lund, Sweden
[2]Department of Ecology and Environmental Science, Umeå University, Umeå, Sweden
[3]Department of Earth and Atmospheric Sciences and School of Biological Sciences, University of Nebraska–Lincoln, Lincoln, Nebraska, USA
[4]Earth and Life Institute, Université catholique de Louvain, Louvain-la-Neuve, Belgium
[5]Institute of Geology and Palaeontology, Faculty of Science, Charles University, Prague, Czech Republic

**Correspondence:** Petra Zahajská (petra.zahajska@geol.lu.se)

**Abstract.** High biogenic silica (BSi) concentration occurs sporadically in lake sediments throughout the world, however, the processes leading to high BSi concentrations varies. While BSi formation and preservation is expected to occur in silica-rich environments with high dissolved silicon (DSi) concentrations such as volcanic and hydrothermal inputs, the factors and mechanisms explaining high DSi and BSi concentrations in lakes remain unclear. We explored the factors responsible

for the high BSi concentration in sediments of a small, high-latitude subarctic lake (Lake 850). To do this, we combined measurements of variations in stream discharges, DSi concentrations and stable Si isotopes in both lake and stream water with measurements of BSi content in lake sediments. Water, radon, and Si mass balances revealed the importance of groundwater discharge as a main source of DSi to the lake, with groundwater-derived DSi inputs 3 times higher than those from ephemeral stream inlets. After including all external DSi sources (i.e., inlets and groundwater discharge) and estimating the total BSi

accumulation in the sediment, we show that diatom production consumes up to 79% of total DSi input. Additionally, low sediment accumulation rates were observed based on the dated core. Our findings thus demonstrate that groundwater discharge and low mass accumulation rate can account for the high BSi accumulation during the last 150 cal. yr BP. Globally, lakes have been estimated to retain one fifth of the annual DSi delivery into the ocean. Well constrained lake mass balances, such as presented here, bring clarity to those estimates of the terrestrial Si cycle sinks.

## 1 Introduction

Diatoms – unicellular golden-brown algae – are found worldwide in all aquatic environments, wetlands, and soils (Battarbee et al., 2002; Clarke, 2003). Diatoms take up dissolved silicic acid $H_4SiO_4$, expresses here as dissolved silicon (DSi), and build their shells in the form of amorphous silica, also known as biogenic silica (BSi). Diatom production is thus a crucial component in the global Si cycle (Tréguer and De La Rocha, 2013). Massive accumulations of fossil diatom frustules in sediments have

been observed in multiple lakes situated in silicon-rich environments, especially on volcanic bedrock, such as Lake Challa, Tanzania/Kenya (Barker et al., 2013), or in hydrothermally active areas, such as Yellowstone Lake, US (Theriot et al., 2006) or Lake Myvatn, Iceland (Opfergelt et al., 2011). However, lakes without volcanism can also accumulate high concentrations of





BSi in the sediment (Frings et al., 2014). One example is the high-elevation and high-latitude lakes, where BSi concentrations as high as 60 weight percent of $SiO_2$ have been found (Frings et al., 2014; Rosén et al., 2010). In addition, large BSi concentrations in sediment have been observed in Lough Neagh, Northern Ireland (Plunkett et al., 2004), Lake Baikal (Swann and Mackay, 2006), Lake Edward (Russell and Johnson, 2005), and Lake Malawi (Johnson et al., 2011). The processes responsible for the diatom-rich sediment formation in these non-volcanic settings, however, are poorly understood.

High BSi accumulation in sediment has been hypothesized to require sufficient DSi concentration in the water column for diatoms to grow and low detrital input to minimize dilution of autochthonous BSi (Conger, 1942). DSi originates ultimately from weathering of bedrock, and it is transported by rivers through the environment where it can be taken up by biological or physical processes (Jenny, 1941). DSi concentrations in the environment are influenced by factors, such as vegetation type (Jenny, 1941; Leng et al., 2009; Struyf et al., 2010), bedrock type (Jenny, 1941; Opfergelt and Delmelle, 2012), indirectly by climate forcing (Fortin and Gajewski, 2009; Jenny, 1941), or watershed geomorphology (Jenny, 1941). In particular, attention has been paid to the relative importance of groundwater discharge as a main source of DSi for a few lakes, such as Lake O'Hara, British Columbia (Hood et al., 2006), Lake Myvatn, Iceland (Opfergelt et al., 2011), Crystal Lake, Wisconsin (Kenoyer and Anderson, 1989; Hurley et al., 1985), at the mouth of the Changjiang river system, China (Zhang et al., 2020), and in Canadian and Siberian rivers (Maavara et al., 2018; Pokrovsky et al., 2013). However, the significance of groundwater discharge is still often overlooked in studies about Si dynamics in lakes.

The contribution of groundwater to lake Si cycle can be evaluated using Si isotopes. Stable Si isotopes are used to trace variation in DSi sources or diatom production and discern processes affecting BSi accumulation in lake sediments. Among the three stable isotopes ($^{28}Si$, $^{29}Si$ and $^{29}Si$) diatoms preferentially take up the lighter $^{28}Si$ (De La Rocha et al., 1997). Diatoms tend to fractionate the Si isotopes with a fractionation factor of $-1.1$ ‰ (De La Rocha et al., 1997), which means that the diatom BSi will have an isotopically lighter signature compared to the source DSi. Riverine DSi usually shows isotopically heavier signatures compared to groundwater, as there are more processes that fractionate Si isotopes during river transport (Frings et al., 2016; Opfergelt and Delmelle, 2012; Sutton et al., 2018). Therefore, the stable Si isotopes provide an ideal tracer for the contribution of groundwater.

Here, we investigate the mechanisms responsible for the diatom-rich sediment formation in high-latitude lake from a non-volcanic setting. Lake 850, northernmost Sweden, is an ideal case study with a high content of BSi in the sediment ca. 40 weight percent (wt%) (Rosén et al., 2010). Oxygen isotopes from diatoms suggested that the lake's isotopic signature is mostly influenced by summer precipitation and variations in the ephemeral inlet streams (Shemesh et al., 2001). Unlike previous studies in this lake, we hypothesize that groundwater discharge is an important mechanism controlling lake DSi concentrations. To test this hypothesis, we estimate groundwater flows discharging into the lake using a water and a radon ($^{222}Rn$) mass balance. DSi concentration and stable Si isotope mass balances were used to determine Si sources for diatom-rich sediment in recent decades.



## 2 Study area

Lake 850 (68°15′ N, 19°7′ E) is located $14\,\mathrm{km}$ southeast from Abisko Research Station (388 m a.s.l.), northern Sweden. From 1913 to 2019, the mean annual temperature was $-0.4\,°\mathrm{C}$, whereas during the study years (2018-2019) the mean annual temperature was $0.03\,°\mathrm{C}$. Further, the mean temperature during the growing season in 2018-2019 (June to August) was $-10.1\,°\mathrm{C}$ (1SD = $2.8\,°\mathrm{C}$), and the long-term (1913-2019) mean summer temperature of $9.8\,°\mathrm{C}$ (1SD = $3.6\,°\mathrm{C}$) (ANS, 2020a). Lake 850 lies at 850 m a.s.l., above the tree-limit, which is at 600 m a.s.l. The lake surface area is $0.02\,\mathrm{km}^2$, with a maximum depth of $8\,\mathrm{m}$ and a catchment area of $0.35\,\mathrm{km}^2$ (Rubensdotter and Rosqvist, 2003). Lake deep basin represents $48\%$ of the lake surface area. The underlying bedrock is composed of granites and syenites and is overlain by a thin layer of till. The catchment vegetation is comprised of Arctic species of mosses, grasses, and shrubs (Shemesh et al., 2001). There are two ephemeral inlets (max $6\,\mathrm{cm}$ deep) in the eastern part of the lake and one outlet ($10\,\mathrm{cm}$ deep) in the western part (Figure 1, Table S1). Besides streams, additional sources of water to rivers and lakes can be snow patches or inputs of groundwater (Pienitz et al., 2008). From mid-October until late May–early June, the lake is ice-covered. The catchment is snow-covered from mid-September to mid-June. In August, the lake is well-mixed, with no thermal stratification. The lake is classified as oligotrophic and has a pH of 6.8 and a dissolved organic carbon concentration of $2.3\,\mathrm{mg\,l}^{-1}$ (Shemesh et al., 2001).

## 3 Numerical analyses – mass balance models

Mass balances for water and radon were constructed to estimate the potential contribution of groundwater discharge to Lake 850. A Si mass balance was used to constrain DSi inputs (inlets, groundwater) and DSi outputs (outlet DSi, sediment BSi accumulation).

### 3.1 Radon mass balance

Radon ($^{222}$Rn, hereafter Rn) is produced from the radioactive decay of $^{226}$Ra (Ra hereafter) present in rocks, soils, and sediments. Radon emanates from Ra bearing minerals, enters the groundwater, and is transported through the aquifer. Groundwaters usually contain Rn concentrations orders of magnitude higher than surface waters, and groundwater discharging into surface waters can thus be easily detected by a Rn enrichment with respect to surface waters (Burnett and Dulaiova, 2003).

Groundwater discharge into the Lake 850 was quantified using a Rn mass balance approach and assuming steady-state (Dimova and Burnett, 2011; Dimova et al., 2013). In the study lake, the sources of Rn are the main inlet streams (n = 2), Rn production by dissolved Ra in the water column, Rn diffusion from underlying sediments, and groundwater discharge. Radon losses include radioactive decay, atmospheric evasion, and losses through the outlet streams (n = 1). Losses by recharge into underlying aquifers are considered minor, because the concentration of Rn seeping into sediments is usually much lower that seeping into the lake (Dimova and Burnett, 2011). By evaluating all Rn source and loss terms, the groundwater flow discharging into the lake can be determined using the following equation:

$$Q_{gw}Rn_{gw} + F_{sed}A + \lambda Ra_{lake}V + Q_{in}Rn_{in} = F_{atm}A + \lambda Rn_{lake}V + Q_{out}Rn_{out} \tag{1}$$



where $Q_{gw}$ is the unknown groundwater discharge $[m^3 \, d^{-1}]$; $Q_{in}$ and $Q_{out}$ are the discharge from inlet and outlet streams $[m^3 \, d^{-1}]$, respectively; $Rn_{lake}$ and $Rn_{gw}$ are the concentrations of Rn $[Bq \, m^{-3}]$ in lake water and groundwater, respectively; $Rn_{in}$ and $Rn_{out}$ are the concentrations of Rn $[Bq \, m^{-3}]$ in the main inlet and outlet streams, respectively; $Ra_{lake}$ is the concentration of Ra in lake water column $[Bq \, m^{-3}]$; $F_{sed}$ is the net diffusive flux of Rn per unit area from lake sediments $[Bq \, m^{-2} \, d^{-1}]$; $F_{atm}$ is the loss of Rn to the atmosphere $[Bq \, m^{-2} \, d^{-1}]$; $\lambda$ is the radioactive decay constant of Rn $[d^{-1}]$; and A $[m^2]$ and V $[m^3]$ are the area and volume of the lake, respectively.

The calculation of Rn loss to the atmosphere was based on the empirical equation by MacIntyre et al. (1995):

$$F_{atm} = k(Rn_{lake} - \alpha Rn_{air}) \tag{2}$$

where k is the gas transfer coefficient $[m \, d^{-1}]$ based on an empirical relationship that relates k with wind speed and lake area (Vachon and Prairie, 2013), and $\alpha$ is the air-water partitioning of Rn corrected for salinity and temperature (Schubert et al., 2012).

Groundwater discharges ($Q_{gw}$) were estimated for August and September 2019. For the remainder months, we interpolated the estimated values by assuming two different scenarios of i) constant or ii) variable groundwater inflows over the year (see Appendix A1) for variable groundwater inflows scenario).

## 3.2   Water balance

The lake water balance was calculated from the volumetric water balance equation:

$$\Delta V = Q_{in} + P + Q_{gw} - Q_{out} - E \tag{3}$$

where $\Delta V$ is the change in lake water volume, $Q_{in}$ and $Q_{out}$ are the stream inflow and outflow, respectively, $Q_{gw}$ is the groundwater inflow, P is precipitation, E is evaporation. Monthly summer precipitation of $48 \, mm$ (ANS, 2020a) has been considered to be included in the stream inflow term. Evaporation and precipitation have been shown to only have a small contribution to the lake water balance, and thus they are considered negligible here (Shemesh et al., 2001).

### 3.3   Silicon mass balance

The DSi flux into and from the lake is calculated as $\phi = Q \cdot c$, where Q is discharge $[l \, s^{-1}]$ and c is DSi concentration $[mg \, SiO_2 \, l^{-1}]$. The DSi balance is then calculated as:

$$\Delta DSi = \phi_{in} + \phi_{gw} - \phi_{out} - \phi_{BSi} \tag{4}$$

where $\Delta DSi$ is the change of lake DSi $[mg \, SiO_2 \, yr^{-1}]$, and $\phi_{in}$, $\phi_{out}$ and $\phi_{gw}$ are the DSi fluxes of the inlet, outlet, and groundwater discharge $[mg \, yr^{-1}]$, respectively. Finally, $\phi_{BSi}$ represents the flux of BSi into the sediment $[mg \, SiO_2 \, yr^{-1}]$, and it was calculated as:

$$\phi_{BSi} = (SAR \cdot \rho_{dry} \cdot BSiwt\% \cdot A_{sed}) \cdot 1000, \tag{5}$$





where SAR is sediment accumulation rate $[\mathrm{cm\,yr^{-1}}]$ calculated from the age-depth model (see Methods section 4.1.2), $\rho_{\mathrm{dry}}$ is dry bulk sediment density $[\mathrm{g\,cm^{-3}}]$, BSiwt% is the mean of BSi content in sediments, $\mathrm{A_{sed}}$ is the area of sedimentary basin of the lake $[\mathrm{cm^2}]$ and 1000 is unit conversion.

Assuming steady-state ($\Delta\mathrm{DSi}=0$), DSi concentration in groundwater was then calculated by dividing $\phi_{\mathrm{gw}}$ from Equation 4 by $\mathrm{Q_{gw}}$. The groundwater DSi flux in ice-free period is dependent on inlet ($\phi_{\mathrm{in}}$), outlet DSi flux ($\phi_{\mathrm{out}}$) and BSi flux to sediment

($\phi_{\mathrm{BSi}}$). However, during ice-covered period, the $\phi_{\mathrm{gw}}$ is dependent only on $\phi_{\mathrm{BSi}}$, if there is some (scenario 1, Appendix B) and on differences of lake volume and DSi concentration. Thus, in order to solve Equation 4, $\phi_{\mathrm{BSi}}$ and lake DSi concentration changes in ice-covered period are required. The $\phi_{\mathrm{gw}}$ during ice covered period is calculated by a mixing model (see Appendix A2).

To constrain DSi concentrations in groundwater, we have examined 3 different scenarios considering different BSi fluxes

($\phi_{\mathrm{BSi}}$) to the sediment driven by the length of diatom production. Two scenarios with maximal and minimal monthly BSi flux appear in the Appendix B aiming to describe maximal and minimal diatom production period and thus groundwater DSi concentrations. The scenario better describing recent diatom production considers that the diatom growing season and, thus, the BSi flux occurs in 4 months, from June until August, in a year (Shemesh et al., 2001), and that scenario is presented here.

### 3.4  Silicon isotope mass balance

The variability of the isotopic Si signature of the lake water is likely to be biologically driven and, therefore, was described using a Si isotopic fractionation model. We hypothesize that the lake has sufficient inlet and groundwater supply to allow for DSi concentrations to remain high and that DSi is unlimited for diatom growth, thus, an open system model was used. The open system model (Varela et al., 2004) describes the expected diatom $\delta^{30}\mathrm{Si_{BSi}}$, as well as the post-uptake signature of the lake water $\delta^{30}\mathrm{Si_{postuptake}}$.

$$\delta^{30}\mathrm{Si_{BSi}} = \delta^{30}\mathrm{Si_{initial}} + \varepsilon \cdot \mathrm{f} \tag{6}$$

$$\delta^{30}\mathrm{Si_{postuptake}} = \delta^{30}\mathrm{Si_{initial}} - \varepsilon \cdot (1-\mathrm{f}) \tag{7}$$

where $\delta^{30}\mathrm{Si_{initial}}$ is the isotopic signature of the initial DSi source, $\varepsilon$ is the fractionation factor of freshwater diatoms ˘$1.1\pm0.41$ ‰ (De La Rocha et al., 1997), and f is the fraction of remaining DSi calculated as $\mathrm{f} = \frac{\mathrm{c_{out}}}{\mathrm{c_{initial}}}$, where $\mathrm{c_{initial}}$ and $\mathrm{c_{out}}$ are DSi concentrations before and after diatom production uptake. Thus, $(1-\mathrm{f})$ is the DSi utilization by diatom production. The initial

DSi concentration is calculated through mixing model with knowledge of the discharges ($\mathrm{Q_{in}}$ and $\mathrm{Q_{gw}}$) and DSi concentrations ($\mathrm{c_{in}}$ and $\mathrm{c_{gw}}$) of the endmembers .

The initial isotopic signature of lake DSi before diatom uptake is back calculated from $\delta^{30}\mathrm{Si_{postuptake}}$ (Appendix A3). The known variables are the $(1-\mathrm{f})$ and the $\delta^{30}\mathrm{Si_{postuptake}}$ represented either in the lake isotopic composition or in the lake outlet $\delta^{30}\mathrm{Si_{out}}$, if $\delta^{30}\mathrm{Si_{lake}} = \delta^{30}\mathrm{Si_{out}}$. Further, the groundwater isotopic composition can be calculated from the initial isotopic Si

mixture before diatom uptake and fractionation through isotope mixing model (see Appendix A3).

Similar to the Si mass balance, the isotope Si mass balance was examined through three scenarios that differ in BSi flux to the sediment representing different length of diatom production (Appendix B). As differences in BSi fluxes alter groundwater





DSi concentrations, the isotopic composition is also changing. However, the scenario describing the recent lake functioning is used for the model presented here. Results of this model were compared with measured data of $\delta^{30}\mathrm{Si}_{\mathrm{BSi}}$ and $\delta^{30}\mathrm{Si}_{\mathrm{postuptake}}$

(which equals to $\delta^{30}\mathrm{Si}_{\mathrm{lake}}$). For validation, the groundwater $\delta^{30}\mathrm{Si}_{\mathrm{gw}}$ for monthly steady-state was calculated and compared with data in the literature.

## 4 Materials and Methods

### 4.1 Sample collection, chemical analyses and chronology

#### 4.1.1 Water sampling

For DSi analyses, water samples from the ephemeral inlets and outlet streams, and lake waters were collected monthly from June to September 2019 (Figure 1, Table S1). Additionally, samples of two profiles of lake water from the deepest and a shallower part of the lake were collected in August and September 2019. All water samples were filtered directly in the field through a $0.45\,\mu\mathrm{m}$ cellulose Sterivex™-HV Durapore filter and acidified with HCl to pH 2 in the laboratory. DSi concentrations were analyzed by the automated molybdate-blue method (Strickland and Parsons, 1972) with a Smartchem 200, AMS

System™ discrete analyzer at Lund University with an instrumental error of $\pm 3.7\%$.

For Rn analyses, surface water samples (maximum of $1.5\,\mathrm{m}$ depth from the surface or $0.5\,\mathrm{m}$ depth at the shallow depths) were collected from 5 different stations (Figure 1, Table S1). A deeper water sample ($4\,\mathrm{m}$ depth) was collected from the central deeper point of the lake to evaluate the potential stratification of Rn concentrations. Samples of water from the main inlet and the outlet stream were also collected. Water samples were collected in $1.5\,\mathrm{l}$ polyethylene terephthalate (PET) bottles with no

headspace using a peristaltic pump. Water was pumped directly into the bottle and left overflowing to replenish the volume at least three times to ensure minimal contact with air. Shortly after collection, Rn concentrations were determined using a Rn-in-air alpha spectrometer RAD7 (Durridge Inc.) coupled to the Big Bottle RAD $\mathrm{H_2O}$ accessory (Durridge Inc.). All Rn concentrations were decay corrected for the time of collection.

Discharges from the inlet and outlet streams were determined by measuring the water velocity at $60\%$ of the sampling point

depth using the six-tenths-depth method (Turnipseed and Sauer, 2010) and creating a cross section through the tributary.

#### 4.1.2 Sediment sampling

Two short ($\sim 15\,\mathrm{cm}$) sediment cores were sampled with a HTH gravity corer in March and August 2019 (Table S1). Both cores showed an undisturbed water-sediment interface. One of the cores was sliced directly in the field in $1\,\mathrm{cm}$ sections. Each section was weighed before and after freeze drying to determine water content, porosity, and wet and dry bulk densities. Total organic

carbon (TOC) and total nitrogen (TN) analyses were carried out on all freeze dried samples, after packing $5$ to $10\,\mathrm{mg}$ of dry sediment into tin capsules. Five samples throughout the core were tested for carbonate content by acidifying with HCl and heating to $60\,°\mathrm{C}$ before the TOC measurements (Brodie et al., 2011). The measurements were done on a COSTECH ECS4010 elemental analyzer at the Department of Geology, Lund University, with the average analytical uncertainty for TOC of $0.3\,\mathrm{wt}\%$



based on duplicate analysis ($n = 14$). The carbonate content calculated as a difference in TOC between de-calcified and bulk
sample was below 0.5 wt%, thus considered negligible.

Biogenic $SiO_2$ content in the sediment was analyzed by sequential alkaline extraction (Conley and Schelske, 2001). Freeze
dried and homogenized samples were digested in $0.1\,M\,Na_2CO_3$ (sample reagent ratio $0.03/40\,g/ml$) in a shaking bath at
$85\,°C$ for 5 hours. Subsamples of $100\,\mu l$ were taken at 3, 4, and 5 hours and neutralized in $9.9\,ml$ of HCl to examine for the
dissolution of minerals. As no changes in the amount of total Si extracted during the time course of the dissolution, the mean
BSi concentration from all the values was used to estimate BSi concentration with no mineral correction applied (Conley,
1998).

All sediment samples were analyzed for radionuclide concentrations ($^{210}$Pb, $^{226}$Ra, and $^{137}$Cs) at Lund University. $^{210}$Pb,
$^{226}$Ra, and $^{137}$Cs were determined by direct $\gamma$-counting using a high-purity germanium detector ORTEC (Model GEM FX8530P4-
RB). Freeze-dried and ground samples were sealed for at least 3 weeks before counting to ensure secular equilibrium of $^{226}$Ra
daughters. $^{210}$Pb was determined through the $46\,keV$ $\gamma$-emission and $^{226}$Ra through the 351 and $609\,keV$ $\gamma$-emission of its
daughter nuclide $^{214}$Pb and $^{214}$Bi, respectively. $^{137}$Cs was measured by its emission at $662\,keV$. Self-absorption was measured
directly, and the detector efficiency was determined by counting a National Institute of Standards and Technology sediment
standard.

Sediment core chronologies were obtained by applying the Bayesian statistics approach with software package Plum (Aquino-
López et al., 2018). The Plum package was applied using the default settings for the thickness of Bacon sections ($1\,cm$). Plum
used the individual $^{226}$Ra measurements as an estimate of the supported $^{210}$Pb concentration. The unsupported $^{210}$Pb was found
in upper most $7\,cm$, and the software package Plum (Aquino-López et al., 2018) extrapolated the ages for the remaining $7\,cm$
based on measured data.

To constrain the Rn mass balance, the second sediment core was used for equilibration experiments in order to determine
Rn diffusion from underlying sediments and the Rn concentration representative of the groundwater discharging into the lake.
Briefly, diffusive flux experiments were carried out in the laboratory by incubating $\sim 200\,g$ of dry sediment placed into $500\,ml$
PET bottles with Milli-Q® water, as described in Chanyotha et al. (2014). Using the RAD7 coupled to the Big Bottle RAD
$H_2O$ accessory (Durridge Inc.), Rn concentrations were monitored for 14 hours. The rate of Rn diffusion from the sediment
($F_{diff}$) was derived from the exponential ingrowth of Rn concentrations with time. The bottles containing grab sediments were
then stored for more than a month and periodically shaken. After this time, the Rn concentration in water was measured using
the RAD7 and converted into groundwater endmember activities using porosity and bulk density as described in Chanyotha
et al. (2014).

## 4.2 Stable Si isotopes analyses

Stable Si isotope analyses were performed on diatoms recovered from sediment, lake, and stream water samples. Cleaned
diatom material from a previous study (Shemesh et al., 2001) was processed for stable Si isotopes. Briefly, pure diatom samples
($\sim 0.8\,mg$) were digested with 0.5 to $1\,ml$ of $0.4\,M\,NaOH$ (analytical purity) at $50\,°C$ for at least 48 hours. When all diatoms
were dissolved, samples were diluted with Milli-Q® water to prevent precipitation and fractionation of amorphous silica, then





neutralized by $0.5$ to $1\,\mathrm{ml}$ of $0.4\,\mathrm{M}$ suprapur® HCl. The solutions were measured for their DSi concentration to obtain the Si recovery, which was between $90$ and $100\%$. Sample solutions were purified for Si isotope analysis by cation-chromatographic

separation using $1.5\,\mathrm{ml}$ cation-exchange DOWEX® 50W-X8 (200-400 mesh) resin following the method of Georg et al. (2006). Silicon from filtered water samples was purified using the same cation-exchange method (Georg et al., 2006). The international Si standard NIST reference material RM-8546 (former NBS-28) and laboratory standard Diatomite were prepared by alkaline NaOH fusion and purified following protocol by Georg et al. (2006).

The reference material RM-8546 (former NBS-28) and laboratory standards IRMM-018, Big-Batch, and Diatomite used in

the VegaCenter were prepared by another type of fusion with $LiBO_2$ (Sun et al., 2010). Thus, our alkaline NaOH fused NBS-28 and Diatomite standards (Georg et al., 2006), purified in identical way as the samples, were matrix matched to contain $3\,\mathrm{mg\,l^{-1}}$ Li IPC-MS standard. Similarly, all purified samples were diluted to a concentration of $3\,\mathrm{mg\,l^{-1}}$ of Si in $0.12\,\mathrm{M}$ SeaStar HCl matrix and doped with Li to contain $3\,\mathrm{mg\,l^{-1}}$ Li to match the standard matrix.

The stable isotope measurements were carried out on a NuPlasma (II) HR multi-collector inductively conducted plasma mass

spectrometry (MC-ICP-MS, Nu Instruments™) with an Apex HF desolvation nebulizer at the Vegacenter, Swedish Museum of Natural History, Stockholm. The $^{28}$Si signal intensity of full procedural blanks was determined to be less than $0.35\%$ of the total signal intensity, thus no sample contamination was observed. Silicon isotope data are reported as deviations of $\frac{^{30}\mathrm{Si}}{^{28}\mathrm{Si}}$ and $\frac{^{29}\mathrm{Si}}{^{28}\mathrm{Si}}$ from the NBS-28 reference solution in ‰, denoted $\delta^{30}$Si and $\delta^{29}$Si as follows:

$$\delta^{30}\mathrm{Si} = \left( \frac{\frac{^{30}\mathrm{Si}}{^{28}\mathrm{Si}\,\mathrm{sample}}}{\frac{^{30}\mathrm{Si}}{^{28}\mathrm{Si}\,\mathrm{NBS28}}} - 1 \right) \cdot 1000. \tag{8}$$

Each sample was measured three times, bracketed by NBS-28 in between, and full chemical replicates for all samples ($n = 25$, total measurements $= 180$) were measured. Secondary reference materials Diatomite, Big-Batch, and IRMM-018 were measured throughout all measuring sessions in a period of 3 years, with averages of $\delta^{30}\mathrm{Si} = 1.26 \pm 0.19\,‰$ ($2\mathrm{SD}_{\mathrm{repeated}}$, $n = 219$) for Diatomite, $\delta^{30}\mathrm{Si} = -10.64 \pm 0.18‰$ ($2\mathrm{SD}_{\mathrm{repeated}}$, $n = 77$) for Big-Batch, and $\delta^{30}\mathrm{Si} = -1.77 \pm 0.18‰$ ($2\mathrm{SD}_{\mathrm{repeated}}$, $n = 100$) for IRMM-018 for quality control purposes. All secondary reference material values were in good agreement with

values from a previous interlaboratory comparison (Reynolds et al., 2007). The reproducibility of all samples was $> 0.2\,‰$. At the Vegacenter laboratory, the long-term precision for $\delta^{30}\mathrm{Si}$ is $0.15‰$ (2SD).

## 5   Results

### 5.1   Lake water chemical and isotopic properties

Lake 850 is a subarctic lake in a region with strong seasonality. The discharge from inlets and the outlet streams show a

decreasing trend throughout the ice-free period from June through September (Table 1). The highest water flow rates are observed during the snowmelt period (June and July). Inflow from the stream inlet to the lake in August is low, and both inlets are dry in September.





During the ice-free period direct surface precipitation contribution from the watershed, was estimated from the average precipitation of $48\,\mathrm{mm\,month^{-1}}$ (ANS, 2020a). With the watershed area of $0.35\,\mathrm{km^2}$ (Rubensdotter and Rosqvist, 2003), precipitation results in $0.65\,\mathrm{l\,s^{-1}}$, which represents only $1.4\%$ of the lake volume. Similar or higher discharges are observed in the stream inlets from July to August. Therefore, the influence of precipitation on the water mass balance is limited. The calculated lake water residence time during the high-flow regime in June, defined as lake volume ($1.2 \cdot 10^5\,\mathrm{m^3}$) divided by the lake outlet discharge (Table 1), is 55 days. During the rest of the year, the lake water residence time is more than 1 year.

Lake DSi concentration varies seasonally (Table 1), with the highest values during the ice-covered period in March, reaching $2.51 \pm 0.35\,\mathrm{mg\,l^{-1}}$, expresses as $SiO_2$. With snowmelt, the lake DSi decreases to $1.24 \pm 0.02\,\mathrm{mg\,l^{-1}}$ in June and to its minimum value of $0.96 \pm 0.06\,\mathrm{mg\,l^{-1}}$ in August. With the first snow in September, lake DSi concentration rebounds, having values of $1.37 \pm 0.04\,\mathrm{mg\,l^{-1}}$. Data of DSi for the inlets and the outlet streams show two different patterns during the year (Table 1). A lower inlet DSi concentration of $2.34 \pm 0.05\,\mathrm{mg\,l^{-1}}$ is observed during snow melt in June compared to July and August, when the inlet DSi concentrations increase to $4.79 \pm 0.05\,\mathrm{mg\,l^{-1}}$ and $5.05 \pm 0.12\,\mathrm{mg\,l^{-1}}$, respectively. The lake outlet DSi concentration shows little variability, with the lowest concentration of $0.94 \pm 0.01\,\mathrm{mg\,l^{-1}}$ in July and only a small increase up to $1.12 \pm 0.03\,\mathrm{mg\,l^{-1}}$ towards the end of the summer season in August. In September, when the inlet streams are snow covered, the DSi concentration in the outlet stream is the same as the lake water concentration at $1.37 \pm 0.01\,\mathrm{mg\,l^{-1}}$.

The stable Si isotope signatures of the lake, inlet, and outlet streams vary during the year. The heaviest lake $\delta^{30}Si_{lake}$ signature, $1.27 \pm 0.15\,‰$, is observed during the ice-cover period, and the lightest signature, $0.73 \pm 0.10\,‰$, occurs during the snowmelt in June (Table 1). In June, the inlet has a lighter $\delta^{30}Si_{in}$ of $0.02 \pm 0.10\,‰$, whereas in August the inlet isotopic signature $0.78 \pm 0.15\,‰$ has similar values as the lake. The $\delta^{30}Si_{out}$ of the outlet in June is slightly heavier ($0.89 \pm 0.10\,‰$) compared to the lake $\delta^{30}Si_{lake}$. In July the outlet $\delta^{30}Si_{out}$ is lighter than the inlet one (Table 1). During the remainder of the year, the outlet $\delta^{30}Si_{out}$ is closely similar to the lake and inlet $\delta^{30}Si_{lake}$.

## 5.2 Groundwater discharge

Surface lake Rn concentrations range between $94\,\mathrm{Bq\,m^{-3}}$ to $136\,\mathrm{Bq\,m^{-3}}$ in August and from $96\,\mathrm{Bq\,m^{-3}}$ to $126\,\mathrm{Bq\,m^{-3}}$ in September. Dissolved Ra in lake waters is assumed to be similar to those found in other lakes in the region ($1.4 \pm 0.6\,\mathrm{Bq\,m^{-3}}$). However, the measured Rn inputs (the stream inlets) due to Ra decay were below $0.5\%$, compared to the net excess of Rn delivered by groundwater discharge. Thus, the inlet Rn flux was neglected in the total Rn balance.

There was no significant vertical stratification of Rn concentration with Rn concentrations in deep waters ($105 \pm 26$ and $79 \pm 24\,\mathrm{Bq\,m^{-3}}$) in August and September, respectively. Equation 1 was solved analytically to obtain the amount of groundwater discharging into the lake ($Q_{gw}$) in August and September 2019. Uncertainties of individual terms were included in the estimation of the associated uncertainty (NORM, 1995; Taylor and Kuyatt, 1994).

Using the average wind-speed for $48\,\mathrm{h}$ period prior sampling ($3.1 \pm 1.2$ and $5.0 \pm 1.8\,\mathrm{m\,s^{-1}}$ in August and September, respectively) resulted in $k_{Rn}$ estimates of $1.1 \pm 0.2$ and $1.2 \pm 0.4\,\mathrm{m\,s^{-1}}$. Uncertainties include the variation of wind speed and uncertainties associated with the empirical equation to estimate $k_{Rn}$. Using the Rn concentration in lake waters, total losses of Rn to the atmosphere are $123 \pm 32$ and $138 \pm 32\,\mathrm{Bq\,m^{-2}\,d^{-1}}$ in August and September, respectively. Radon losses due to





decay were $125 \pm 22$ and $123 \pm 15 \,\mathrm{Bq\,m^{-2}\,d^{-1}}$, respectively, where uncertainties are obtained from the analytical uncertainties for Rn concentrations in lake waters. Losses of Rn through the outlet stream were $7 \pm 4$ and $9 \pm 4 \,\mathrm{Bq\,m^{-2}\,d^{-1}}$. Among all Rn losses, atmospheric evasion (50%) and decay (47%) were the terms that have the largest contribution to the Rn mass balance.

Radon losses through the outlet stream are almost negligible (3%).

Diffusive Rn flux from underlying sediments ($\mathrm{F_{diff}}$) obtained from diffusion experiments in the lab is $89 \pm 17 \,\mathrm{Bq\,m^{-2}\,d^{-1}}$, and it is one of the main sources of Rn into the system. Fluxes of Rn from the sediment compensate for up to 57% of total Rn losses. Uncertainties associated with this flux are from analytical uncertainties in the slope for the regression analyses of the increase in Rn concentration through time in the sediment diffusion experiment. Due to the low concentrations of Ra in lakes

from the same area ($1.4 \pm 0.6 \,\mathrm{Bq\,m^{-3}}$, *C. Olid, unpublished data*), Rn inputs due to Ra decay were considered negligible in the Rn mass balance.

Rn inputs from groundwater are required to balance the Rn losses from the lake. The Rn flux into the lake through groundwater discharge is calculated to be $166 \pm 43 \,\mathrm{Bq\,m^{-2}\,d^{-1}}$ and $180 \pm 40 \,\mathrm{Bq\,m^{-2}\,d^{-1}}$ in August and September, respectively. Considering the lake area of $20\,000 \,\mathrm{m^2}$ and the Rn concentration in groundwater obtained from incubation experiments

($10626 \pm 1720 \,\mathrm{Bq\,m^{-3}}$), groundwater fluxes are $3.56 \pm 1.25 \,\mathrm{l\,s^{-1}}$ and $3.88 \pm 1.06 \,\mathrm{l\,s^{-1}}$ for August and September, respectively. Note that this is a conservative estimate for groundwater fluxes, because we use the highest measured Rn concentration as the endmember.

Due to the lack of Rn measurements for the entire year, we estimated groundwater inputs for the months where no sampling was carried out using two scenarios: (i) constant groundwater inflow of $3.73 \pm 1.25 \,\mathrm{l\,s^{-1}}$, calculated as the mean of the August

and September data; and (ii) modelled groundwater inflow based on groundwater fluxes obtained from a lake survey in the Abisko region in 2018–2019 (*C. Olid, unpublished data*), which ranged from $1.55 \pm 1.09 \,\mathrm{L\,s^{-1}}$ to $11.20 \pm 2.34 \,\mathrm{l\,s^{-1}}$ (Figure 2). The annual Rn fluxes follow a pattern of a distinct peak in discharge in June and a gradual decrease towards July – October, reaching the base-flow level in November (Figure 2). The ratio between the groundwater Rn flux in September in Lake 850 and the groundwater Rn fluxes from the lake survey was used to model the missing groundwater Rn fluxes in Lake 850 (Figure

2, Appendix A1).

### 5.3    Age-depth model, lithology and mass accumualtion rates

The age-depth model for the sediment core is shown in Figure 3. The average sediment accumulation rate (SAR) was estimated to be $0.083 \pm 0.041 \,\mathrm{cm\,yr^{-1}}$, which equals a sediment accumulation rate of $12 \pm 6 \,\mathrm{yr\,cm^{-1}}$ and a mass accumulation rate (MAR) of $16.0 \pm 9.3 \,\mathrm{mg\,cm^{-2}\,yr^{-1}}$. The presence of mosses in the sediment was observed during the core processing and also was

described in the sediment lithology by Shemesh et al. (2001). Changes in the sediment content of aquatic or terrestrial mosses, was also supported by the C/N ratio (Figure 4), suggesting this is the cause of changes in MAR.

Lake 850 sediment is composed of carbonate-free clay gyttja with an average TOC content of 11.4 wt%, average TN of 1.1 wt%, and a resultant C/N ratio of 10.2 (Figure 4). Sediment porosity as high as $89.5\%$ is found in the surface sediment, where sediment dry bulk density average is $0.19 \pm 0.06 \,\mathrm{g\,cm^{-3}}$. The BSi concentration along the sediment varies from $13.2 \pm 0.28$

wt% to $22.8 \pm 0.24$ wt%, with the highest BSi concentration in the surface of the core. The BSi concentrations reported here



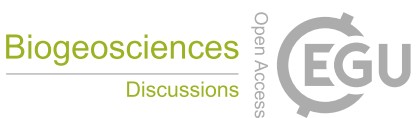

are lower than previous measurements. For example, BSi concentration in the surface sediment of Lake 850 was previously reported to be 40.3 wt% (Rosén et al., 2010), which is twice the value found here, demonstrating the high variability of BSi in the sediments.

Using the average MAR rate and the average BSi wt% of 18.2 wt% we estimated the BSi accumulation rate ($\phi_{BSi}$) to be

$2.9 \pm 1.5 \, \mathrm{mg \, SiO_2 \, cm^{-2} \, yr^{-1}}$. BSi accumulation rates show stable values around $1.8 \, \mathrm{mg \, SiO_2 \, cm^{-2} \, yr^{-1}}$ in the upper $7 \, \mathrm{cm}$ of the core, whereas an increase in BSi accumulation is observed towards the bottom $7 \, \mathrm{cm}$ of the core (Figure 4) likely connected to the higher MAR. The average diatom isotopic signature ($\delta^{30}\mathrm{Si_{BSi}}$) measured on cleaned diatoms from the upper most $8 \, \mathrm{cm}$ of sediment core from 1999 (Shemesh et al., 2001) is $0.07 \pm 0.43 \, \%o$ (n = 3).

## 6  Mass balance models

### 6.1  Water balance

Two water balance scenarios were considered where changes in the lake level were evaluated: (i) constant groundwater inflow over the annual cycle as an additional water source, and (ii) modelled groundwater discharges varying throughout an annual cycle (Figure 5 and A1). In both scenarios, lake-level increases during the ice-covered period (Figure 5, blue and green line) are a result of a potential groundwater inflow. This accumulated water is released through the outlet when the lake ice starts

to melt in May-June, and the outlet discharge is thus high (Table 1). After this period, lake-level is stabilized and groundwater replenishes the lake original volume during short periods over the summer.

When groundwater discharge is assumed to be constant (Scenario i, $3.73 \pm 1.25 \, \mathrm{L \, s^{-1}}$) based on our data from August and September, the lake shows annual lake-level changes up to $1.9 \, \mathrm{m}$ (Figure 5 and A1, blue line). From July to December, the lake volume is restored by the groundwater inflow, and, on the annual time scale, the lake-level would increase around $2 \, \mathrm{m}$ every

year (Figure 5, blue line).

Using the modelled annual groundwater inflow (Scenario ii, Figure 2), limited lake-level changes were observed. The maximum lake-level decrease is $95 \, \mathrm{cm}$ during summer (Figure 5 and A1, green line), but groundwater discharge restores lake-level during upcoming months. Taking into account the uncertainties, lake-level variation can be as great as $2.4 \, \mathrm{m}$ or none (Figure 5 and A1, green shading). This scenario with the smallest lake-level changes is in agreement with previous results of oxygen

isotopes mass balance (Shemesh et al., 2001). Therefore, we used this water balance model further for the Si balances.

### 6.2  Silicon and Silicon isotope mass balance

BSi accumulation occurs in conditions when the total DSi influx is higher that the stream DSi outflux. Therefore, we construct a Si mass balance based on stream inlets and the outlet. The DSi influx through the inlet stream is not sufficient to maintain lake DSi concentration at steady-state in June (red and blue triangles, Figure 6A). In contrast, in July and August sufficient DSi

enters the lake to supply the outlet DSi flux. The monthly inlet DSi flux is between $0.22 \pm 0.11$ to $0.62 \pm 0.31 \, \mathrm{kg \, SiO_2 \, day^{-1}}$, while the outlet DSi flux ranges from $0.19 \pm 0.10$ to $2.21 \pm 1.11 \, \mathrm{kg \, SiO_2 \, day^{-1}}$. However, diatom production is an additional





sink of Si by creating a BSi flux into the sediment. The DSi influx is, thus, not sufficient to account for both the DSi outflux and the BSi flux into the sediment (Figure 6A). Therefore, an additional external source (i.e., groundwater discharge) must supply additional DSi to compensate for the average BSi flux ($2.9\,\text{mg SiO}_2\,\text{cm}^{-2}\,\text{yr}^{-1}$) into the sediment.

Groundwater discharges from scenario ii (Figure 2) were used to build a Si mass balance and a Si isotope mass balance. Here, we assume that the recent BSi flux into the sediment occurs only during the growing season (from June until September) (Figure 6A; Shemesh et al., 2001). The missing DSi flux resulting from the mass balance was considered to originate from the groundwater flux, and thus, we use this flux to calculate back the groundwater DSi concentration and isotopic signature.

During the diatom growing season, BSi flux into the sediment increases up to $1.76 \pm 0.87\,\text{kg SiO}_2\,\text{day}^{-1}$ (magenta line,
Figure 6A), which produces DSi deficiency in the lake. To balance this deficiency, groundwater discharge must supply between $1.62 \pm 1.21$ and $3.39 \pm 1.77\,\text{kg SiO}_2\,\text{day}^{-1}$ during the diatom growing season (cyan line, Figure 6A). Considering the modelled groundwater discharges derived from Rn mass balance, the DSi concentration in the groundwater is estimated to range from $3.50 \pm 1.68\,\text{mg l}^{-1}$ to $5.85 \pm 2.99\,\text{mg l}^{-1}$ from diatom growth (cyan line, Figure 6B). During the ice-covered period, the diatom growth ans thus BSi flux into the sediment is considered to be negligible, while groundwater is still flowing into the lake.
The winter groundwater concentration is calculated from the difference in the lake concentration from September ($1.02 \pm 0.91\,\text{mg l}^{-1}$) to March ($2.51 \pm 0.35\,\text{mg l}^{-1}$) (Appendix A2). Therefore, the groundwater discharging into the lake from late-October until mid-June is the only water inflow with a DSi concentration of $6.95 \pm 4.90\,\text{mg l}^{-1}$.

The Si isotopes mass balance using the open fractionation model (Varela et al., 2004) shows that the higher demand of DSi in the productive months (Figure 6A, B) needs to have a lighter isotopic composition in order to produce the $\delta^{30}\text{Si}_{\text{BSi}}$ of
$0.07 \pm 0.43\text{‰}$ measured on diatoms preserved in the sediment. The isotopically lighter source is assumed to be groundwater discharge, with calculated ranges from $-0.55 \pm 0.55\text{‰}$ in July to $0.23 \pm 0.58\text{‰}$ in September (Figure 6C). Using the modelled groundwater $\delta^{30}\text{Si}$, the expected $\delta^{30}\text{Si}_{\text{BSi}}$ in all productive months varies from $-0.49 \pm 0.49\text{‰}$ to $-0.01 \pm 0.56\text{‰}$ (not shown), values that are in agreement with the sediment BSi of $\delta^{30}\text{Si}_{\text{BSi}} = 0.07 \pm 0.43\text{‰}$. The production consumes from 63% of the initial DSi in June, 77% in July and September, and 79% in August. During the ice-covered period from late-October until
mid-June, the groundwater base flow is considered to be constant, calculated from the difference of the lake isotopic signatures from September until March (Appendix A3), and thus the $\delta^{30}\text{Si}_{\text{gw}} = 1.45 \pm 2.58\text{‰}$ (Figure 6C).

## 7    Discussion

Lake 850 is unusual in terms of both the DSi and BSi concentration in water and sediment, respectively. The maximum DSi concentration of $2.51\,\text{mg SiO}_2\,\text{l}^{-1}$ in March is among the top 10% of lakes in Northern Sweden (Bigler and Hall, 2002). The
average BSi content in the lake sediment of 40 wt% (Rosén et al., 2010) places Lake 850 in the upper 6% of lake sediments studied worldwide (Frings et al., 2014). Although several factors, including the morphology of the watershed (Jenny, 1941; Rubensdotter and Rosqvist, 2003), diatom production and low detrital input (Conger, 1942), vegetation (Struyf et al., 2010), and preservation potential (Ryves et al., 2003) are known to affect sedimentation regimes and BSi accumulation resulting in





a diatom-rich sediment, we show here that groundwater input is an important factor leading to the large BSi accumulation in
Lake 850.

The combined results from the water, Rn, and Si mass balances indicated the importance of an external source of DSi
through groundwater discharge. Groundwater inflow was the primary water and DSi supply to the lake, with a contribution
about 3 times higher than the stream inlets (Figure 6A). The Si and Si isotope mass balance models showed that groundwater
DSi concentration and isotopic composition varied during the ice-free period, compared to the ice-covered period, when they
were stable (Figure 6B, C).

The significance of groundwater on lake Si cycle is also evidenced by the relatively lighter stable Si isotope signature of
diatoms from sediment, which suggests that groundwater is the primary DSi source for diatoms. Stream inputs could also be
a source of DSi for diatoms, especially in early spring, when snowmelt can deliver isotopically lighter DSi by displacement
of shallow groundwater into the stream inlet (Campbell et al., 1995). However, spring snowmelt water and groundwater in
June are likely to have the same isotopic composition (Figure 6C) because the same factors, e.g., short residence time in
the watershed are present in both types of water. Thus, only by using mass balance is the quantification of each DSi source
apparent, providing evidence that groundwater supplies almost 4 times more DSi compared to streamflow in June. Our results
suggest that the groundwater supply plays a crucial role in providing DSi for the production of diatoms and accumulation of
BSi in Lake 850.

## 7.1 The role of groundwater in the water balance

The water balance coupled with the Rn mass balance indicated that groundwater discharge is an essential water source for the
lake. Both models of groundwater inflow (constant and varying groundwater inputs) demonstrated changes in lake volume as a
result of high-water discharge at the outlet of the lake during spring snowmelt. More pronounced changes in lake volume were
observed in scenario i, where constant groundwater inflow was assumed (Figure 5, blue line). However, because the oxygen
isotope data showed negligible evaporation and precipitation effect on lake volume change (Shemesh et al., 2001), this model is
not considered to be the most realistic. Scenario ii, which considered a variable groundwater flow (Figure 5, green line) seems
to be more realistic. The modelled groundwater hydrograph (Figure 2) is comparable with the hydrograph of the neighbouring
river Miellejokha (Figure S1) and resembles the hydrographs of groundwater discharge in studies of high-altitude lakes from
other regions (Clow et al., 2003; Hood et al., 2006; Huth et al., 2004; Liu et al., 2004). The results from this model show that
groundwater discharge is up to 5 times higher values than the lake water outflow through the outlet. Similarly, groundwater
discharge brings from 3 to 24% of the lake volume depending on the month.

The water balance based on modelled groundwater inflow suggests that lake-level changes throughout the year are within
a range of 0.95 m (Figure 5, green line), and, thus, lake area and average depth also vary throughout the year. Therefore, the
underlying assumptions of constant depth and area are likely overestimating lake-level change. For a more precise model of
lake-level, lake volume variations and a detailed bathymetry of Lake 850 is needed. However, the importance of the ground-
water contribution to Lake 850 supports the evidence that groundwater should be considered as an important water and DSi





source for high-altitude and high-latitude lakes, with support of data on groundwater DSi in Lake O'Hara (Hood et al., 2006), Lake Myvatn (Opfergelt et al., 2011) and Crystal Lake (Hurley et al., 1985; Kenoyer and Anderson, 1989).

### 7.2 The role of groundwater in Si concentration mass balance and Si isotope mass balance

The lake Si mass balance (Figure 6A) shows that modelled groundwater concentration and flux of BSi vary through the year, which is similar to observations from Crystal Lake in Wisconsin (Hurley et al., 1985). Seasonal variations in groundwater DSi concentration related to discharges were also observed in Canadian rivers with groundwater inputs (Maavara et al., 2018). Moreover, the calculated BSi flux into the sediment is comparable (or higher) with BSi fluxes observed in some of the North American Great Lakes (Conley, 1988; Newberry and Schelske, 1986; Schelske, 1985) and lakes with diatomaceous sediment

in the Arctic (McKay et al., 2008; Kaplan et al., 2002; Tallberg et al., 2015).

The model of stable Si isotopes shows little variation during the ice-covered period, as no diatom production is expected. The $\delta^{30}\mathrm{Si}$ of groundwater for the ice-covered period (Figure 6C) falls into the range of measured groundwater isotopic composition worldwide, which ranges from $-1.5$ to $2‰$ (Frings et al., 2016). However, the groundwater signature $\delta^{30}\mathrm{Si_{gw}}$ is heavier than found in other groundwater studies (Georg et al., 2009; Opfergelt et al., 2011; Ziegler et al., 2005), which may reflect

lower dissolution of primary minerals, longer groundwater residence time, and possibly some clay mineral formation in the groundwater pathway (Frings et al., 2016; Pokrovsky et al., 2013) during the ice-covered period. Further, no diatom production, and thus no associated Si isotope fractionation, is expected in winter. Therefore, the $\delta^{30}\mathrm{Si_{lake}}$ is influenced by the input of $\delta^{30}\mathrm{Si_{gw}}$ only and not by diatom production. The $\delta^{30}\mathrm{Si_{lake}}$ measured in March is slightly lighter than all modelled $\delta^{30}\mathrm{Si_{gw}}$ for the ice-covered period, which can be explained by diatom dissolution in the uppermost sediment layers. However, if the

uncertainties of the modelled groundwater isotopic composition are taken into account, the lake signature is within the same range as the groundwater signature. Therefore, no additional processes must be present during the ice-covered period, and the groundwater isotopic signature is reflected in the lake isotopic signal. With snowmelt, the decrease of the $\delta^{30}\mathrm{Si_{gw}}$ reflects the increase in weathering of primary minerals and decrease in the groundwater residence time due to higher discharges, as also observed in Arctic rivers (Pokrovsky et al., 2013).

The greatest variation in the isotopic signature of groundwater occurs in August, when the groundwater isotopic composition is fully dependent on the changes in BSi flux into the sediment. As the yearly BSi accumulation occurs during the growing season which is only 4 month, the groundwater must bring additional DSi to supply diatom production. Hence, the isotopic model calculating the groundwater isotopic composition shows $\delta^{30}\mathrm{Si_{gw}}$ comparable with values for groundwater reported in the small number of other studies (Frings et al., 2016; Opfergelt et al., 2011). Further, the calculated $\delta^{30}\mathrm{Si_{BSi}}$ based on the

initial mixture of the modelled groundwater and stream inlet signature gives results within the range of the measured $\delta^{30}\mathrm{Si_{BSi}}$.

### 7.3 Model uncertainties

The largest sources of uncertainty in the water and silicon balance models (Figure 5, SA1 and 6) are the discharge uncertainties of the inlet and outlet and the winter groundwater discharges. The spring snowmelt is dynamically changing the inlet and outlet discharges, as has been observed on rivers in the area, such as Miellejohka (Figure S1). With only a single sample



every month, there is no information on variation of the stream on a finer temporal scale. Thus, monthly stream flow and the modelled groundwater discharges might be over- or underestimated. Further, the uncertainties in isotopic model and the isotopic composition of the groundwater were propagated from the mass balance model and from the stable isotopic measurements, especially in the outlet water in August.

Another source of uncertainties in the Si and Si isotope mass balance models originates from the uncertainties on the age-
depth model. The uncertainties on MAR, which are calculated from the SAR and the densities are as high as 50%. It is likely due to changes in the sediment composition and increased content of mosses. Therefore, the BSi flux to the sediment carries similar or higher uncertainty. As a result of those uncertainties, the modelled groundwater concentrations and isotopic composition are ranging greatly.

Additionally, the diatom preservation efficiency, which is globally around $3\%$ in the oceans (Treguer et al., 1995), and, in
deep lakes around $1-2\%$ (Ryves et al., 2003) of the total diatom production, suggests that $97-99\%$ of diatom BSi is redissolved in the water column in those environments. However, no estimates of sediment preservation efficiency are available for small, cold lakes such as Lake 850. Therefore, the mass balance can be slightly underestimated, in case that the BSi flux into the sediment, which was calculated from the sediment record represents only a fraction of the total production. To eliminate this source of uncertainty annual monitoring of diatom production and accumulation would be needed.

Uncertainty also results from the variability among sediment cores in their BSi content. BSi concentrations in the sediment vary from 13 to 40 wt% in different cores (this study; Rosén et al., 2010). We have tested the combination of the MAR ($16.0 \, \mathrm{mg \, cm^{-2} \, yr^{-1}}$) reported from this study with the highest BSi of $40.3$ wt% from a companion core from Lake 850 (Rosén et al., 2010) to evaluate the impact of BSi flux on the groundwater concentrations. The yearly BSi flux would increase 2.2 times, which would result in $1.6$ to $2.3$ times higher groundwater DSi concentration to support the BSi flux and keep the Lake
850 at steady-state. However, the BSi content is variable within the sedimentary basin, and thus the sedimentation rate is a crucial factor for the estimate of BSi accumulation. For future model improvement a monitoring of all inlets, groundwater, pore water, and the outlet together with sediment traps to constrain the production, BSi flux and dissolution would be needed.

## 8 Conclusions

The diatom-rich sediment in Lake 850 is formed because of high DSi supply by groundwater during the growing season for
diatoms coupled with low sedimentation rates, which fosters a large accumulation of diatoms in the form of BSi. Water and Si mass balance demonstrated the importance of groundwater as a source of water and DSi, with fluxes that are 3 times greater than stream input. Groundwater supplies lighter $\delta^{30}$Si, which is reflected in the lighter diatom $\delta^{30}$Si signature. By quantifying the groundwater inputs, the Si and Si isotopic mass balances allowed for the estimate of the stable Si isotope signatures of groundwater throughout the year. The modelled isotopic signature of groundwater falls into the same range as the world
groundwater $\delta^{30}$Si signature (Frings et al., 2016; Sutton et al., 2018).

The results from our study can be applied more broadly to other lakes to evaluate factors governing the accumulation of diatom-rich sediment. BSi rich sediments are likely to be found in lakes situated on silica-rich bedrock, such as in Lake





Challa, Tanzania/Kenya (Barker et al., 2013) or as shown here in lakes with sufficient DSi inputs from groundwater source that supply DSi during the growing season to alleviate potential DSi limitation of diatom growth. In addition, lakes with high

autochthonous carbon production and deposition combined with very low mean sedimentation rates generally found in Arctic lake sediments (Wolfe et al., 2004), as well as lakes with low-relief watershed morphology and with low stream input that yield low quantities of fine-grain clastic input, are potential systems for high BSi accumulation (Conger, 1942). These water bodies with high BSi accumulation act as important sinks of Si in the global Si cycle. Our results support the importance of groundwater in the lake silicon budget and suggest that this process should not be overlooked in future investigations on BSi

in lakes and global estimates of the terrestrial lake BSi sink.





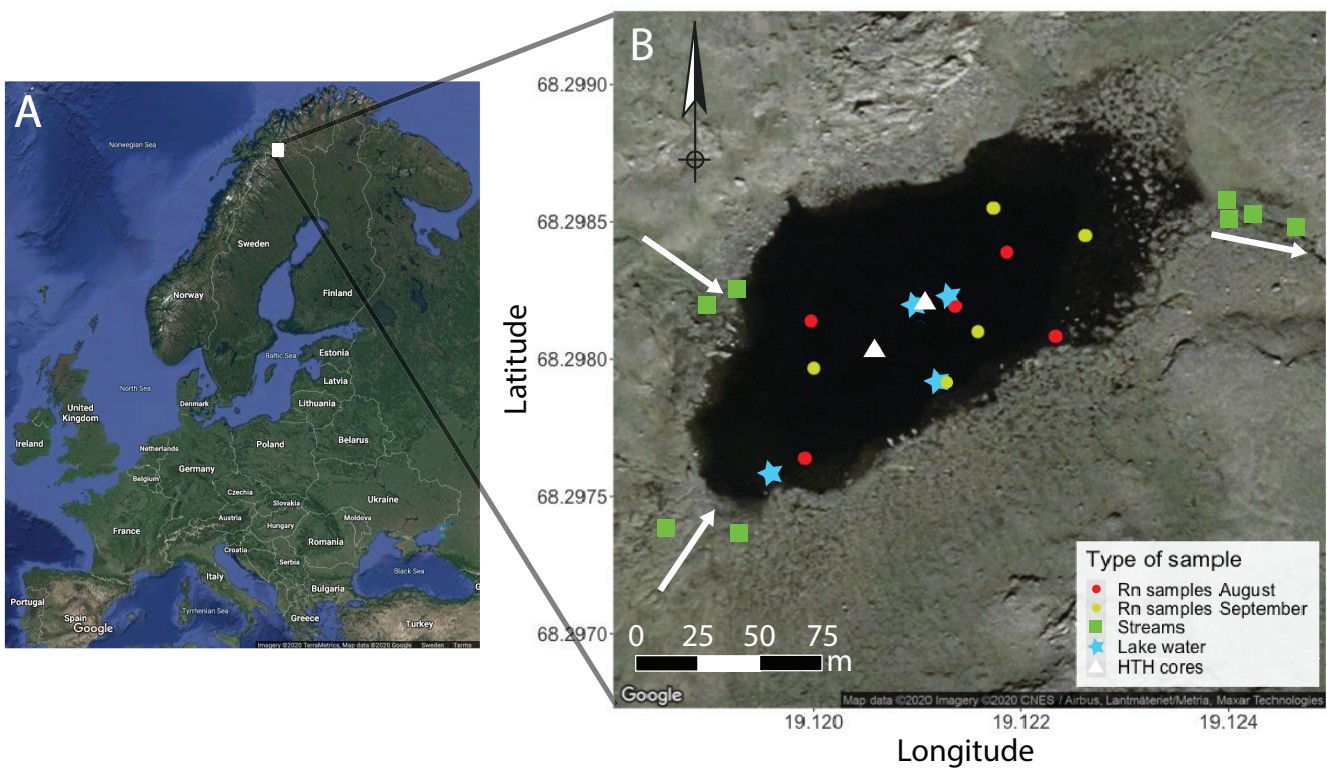

**Figure 1.** Sampling sites of Lake 850 (northern Sweden). Inlets and outlet streams are signified by white arrows. Plotted in R using package ggmaps (Kahle and Wickham, 2013) and modified.





**Table 1.** Summary of discharge from the inlets ($Q_{in}$), the outlet ($Q_{out}$) stream, groundwater discharge ($Q_{gw}$), dissolved Si concentration as $mgSiO_2 l^{-1}$ in the inlets ($c_{in}$), the outlet ($c_{out}$), and the lake water ($c_{lake}$), stable Si isotopic signal of the inlet ($\delta^{30}Si_{in}$), outlet ($\delta^{30}Si_{out}$), and the lake ($\delta^{30}Si_{lake}$).

| | | March | June | July | August | September |
|---|---|---|---|---|---|---|
| $Q_{in}$ | $L \cdot s^{-1}$ | | 2.9 | 1.5 | 0.5 | dry |
| $Q_{out}$ | $L \cdot s^{-1}$ | | 21.5 | 9.9 | 4.5 | 1.6 |
| $Q_{gw}$ | $L \cdot s^{-1}$ | not sampled | not sampled | not sampled | $3.57 \pm 1.24$ | $3.88 \pm 1.06$ |
| $c_{in}$ | $mg \cdot L^{-1}$ | | $2.34 \pm 0.05$ | $4.79 \pm 0.05$ | $5.05 \pm 0.12$ | dry |
| $c_{out}$ | $mg \cdot L^{-1}$ | | $1.19 \pm 0.02$ | $0.94 \pm 0.01$ | $1.12 \pm 0.03$ | $1.37 \pm 0.01$ |
| $c_{lake}$ | $mg \cdot L^{-1}$ | $2.51 \pm 0.35$ | $1.24 \pm 0.02$ | not sampled | $0.96 \pm 0.06$ | $1.37 \pm 0.04$ |
| $\delta^{30}Si_{in}$ | ‰ | | $0.02 \pm 0.10$ | $0.72 \pm 0.10$ | $0.78 \pm 0.15$ | dry |
| $\delta^{30}Si_{out}$ | ‰ | | $0.89 \pm 0.10$ | $0.61 \pm 0.10$ | $0.79 \pm 0.12$ | $1.09 \pm 0.20$ |
| $\delta^{30}Si_{lake}$ | ‰ | $1.27 \pm 0.15$ | $0.73 \pm 0.10$ | not sampled | $0.77 \pm 0.32$ | $1.02 \pm 0.24$ |



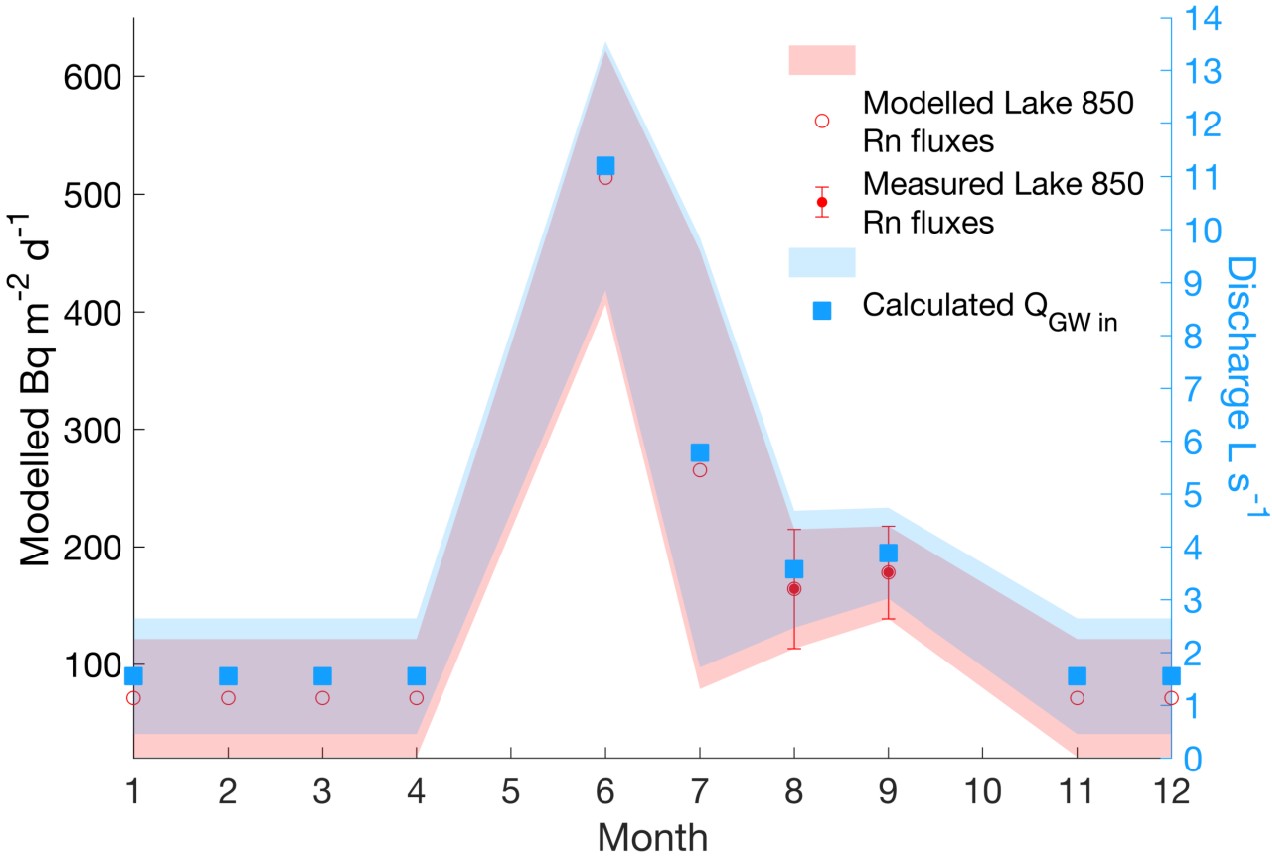

**Figure 2.** The modelled groundwater radon flux of Lake 850 (red circles) based on Rn fluxes in 10 other lakes in Abisko (*Olid et al., unpublished data*), the measured Rn fluxes in August and September (red filled points), and the calculated groundwater discharge in $1\,\mathrm{s}^{-1}$ throughout the year (blue squares). Uncertainties are shown as error bars and with shading.



**Figure 3.** Age-depth model of HTH core. Red line is the median probability age from all age-depth iterations. Grey shading represents age model probability and contains 95% confidence interval (dashed lines). The blue rectangles are the unsupported $^{210}$Pb concentration in $\mathrm{Bq\,kg^{-1}}$. Iteration history (left inset), prior and posterior densities of the mean accumulation rate (second left inset), and prior and posterior of the memory (middle inset), the $^{210}$Pb influx (second right inset), and supported $^{210}$Pb in $\mathrm{Bq\,kg^{-1}}$ (right inset).

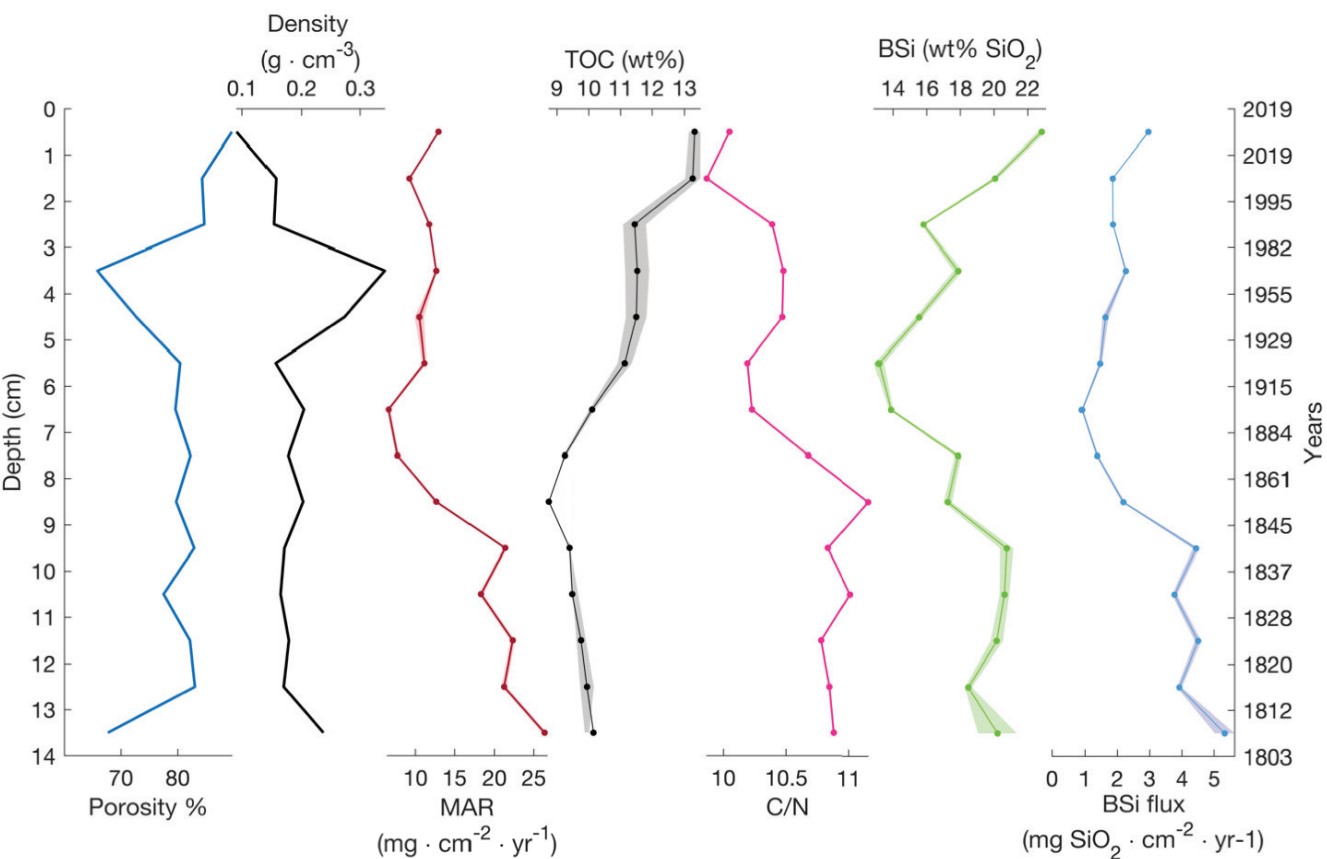

**Figure 4.** HTH core sediment properties (porosity and dry bulk density), mass accumulation rate (MAR) and sediment density. Total organic carbon (TOC) and C/N showing changes in lake carbon content and sources. Biogenic silica (BSi) and BSi flux calculated from MAR and BSi concentrations. One standard deviation is shown by shading.





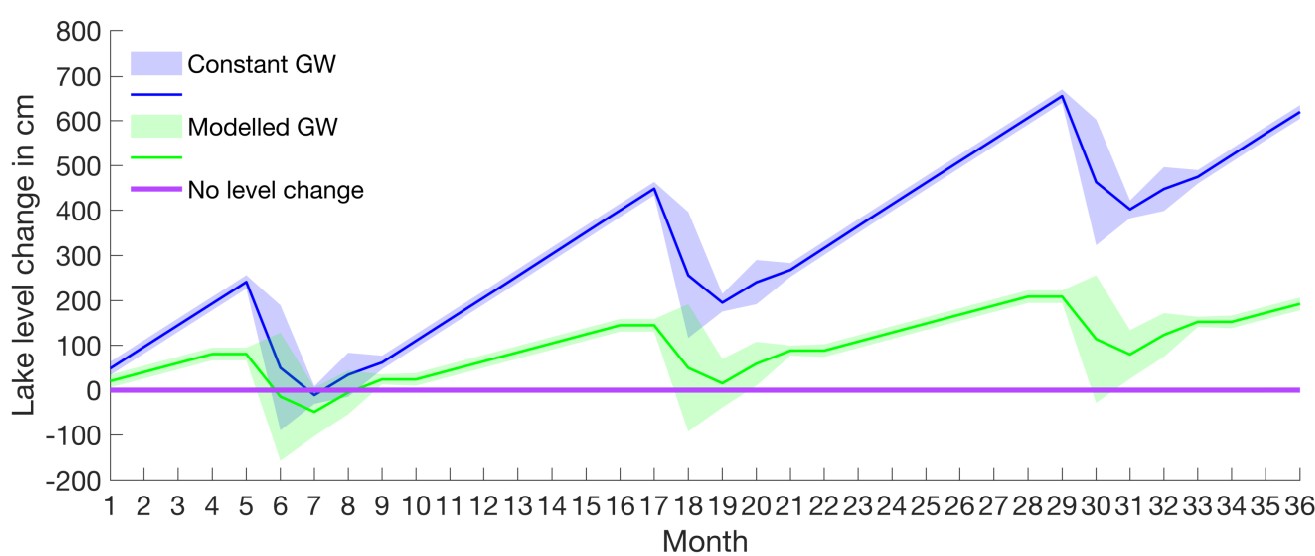

**Figure 5.** Long-term lake-level change calculated based on lake volume changes and water balance. The purple solid line indicates the lake-level starting point. The blue line with shading is the lake-level change with constant groundwater flow (scenario i), and the green line with shading is the lake-level change based on water balance with modelled groundwater discharges (scenario ii).

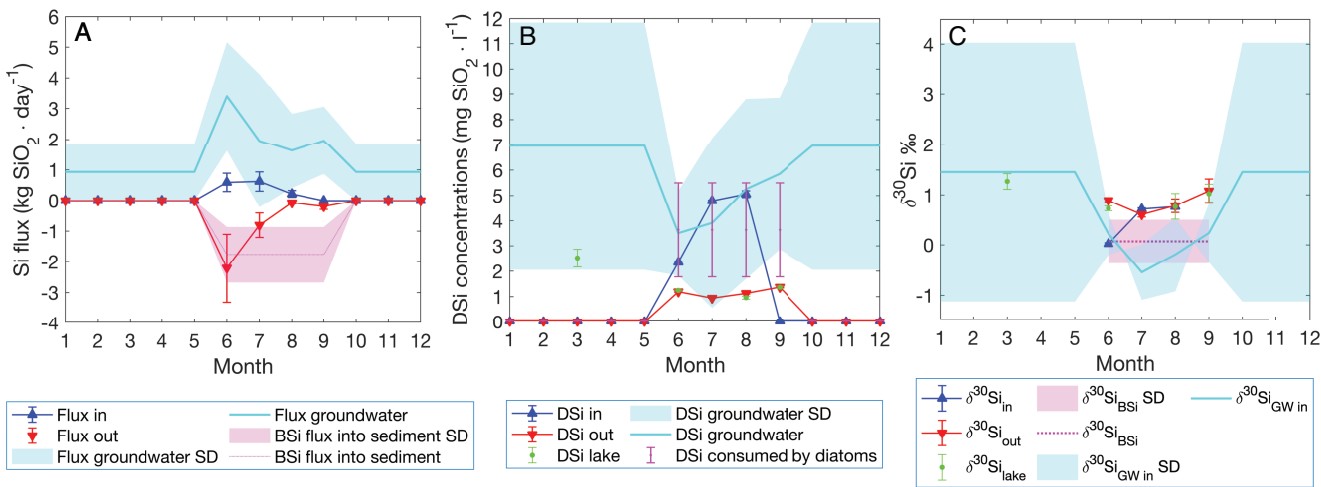

**Figure 6.** Si and Si isotope mass balance model of Lake 850 throughout the year. (A) Mass balance showing the stream DSi influx (blue triangles), the lake outlet DSi outflux as negative flux (red triangles), the diatom BSi flux based on a diatom bloom season lasting 4 months (magenta dotted line), also as a negative flux. The calculated groundwater DSi flux is shown as a positive flux (cyan line). (B) The monthly changes in the DSi concentration of the inlet (blue triangles), outlet (red triangles), lake (green circles), diatom DSi uptake (magenta circles), and groundwater (cyan line). (C) The stable Si mass balance showing monthly variation of the isotopic composition of all DSi sources and sinks. Shading and error bars represent uncertainties.

*Data availability.* All data, if not directly available in tables and appendices, will be available in the PANGAEA database. In the meantime data is available upon request to the authors.

*Author contributions.* PZ and DJC designed the research. PZ and CO carried out the fieldwork. CO performed radon measurements and radon mass balance. PZ performed the TOC, TN, BSi, DSi, stable Si isotopes analyses and performed the data processing. All authors
contributed to discussion and data interpretation. PZ and CO wrote the paper with contributions and comments provided by JS, SCF, SO and DJC.

*Competing interests.* No competing interests are present.

*Disclaimer.* The authors declare that they have no conflict of interest.

*Acknowledgements.* This work was supported by The Royal Physiographic Society in Lund to PZ, the Swedish Research Council to DJC,
and NSF EAR-1514814 to SCF. Part of this work was supported by a FORMAS (d.nr. 2018-01217) grant attributed to CO. We also thank to Aldo Shemesh for providing diatom samples, Christian Bigler, Reiner Gielser and Carl-Magnus Morth for advice and help with fieldwork design. Further we thank the organizations and the individuals who helped with the fieldwork and provided us with equipment and advice: Thomas Westin, Keith W. Larson, Erik Lundin, Svante Zachrisson, CIRC and field assistants Albin Bjärhall, Mathilde, Lukas, Rosine Cartier, Geert Hensgens and Jan Foniok. We acknowledge Hans Schöberg and Melanie Kielman for assistance during sample preparation and isotope
data acquisition. This is Vegacenter contribution number # XXX (number will be provided upon acceptance).



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





**Appendix A: Methods**

**A1    Modelling groundwater Rn fluxes**

Radon fluxes for 10 lakes from the Abisko region were estimated using the same approach in summer and autumn 2018 and 2019 (*C.Olid, unpublished data*). The derived Rn fluxes obtained from the lake survey were used here to model groundwater fluxes through the year in Lake 850. To do this, we divided the estimated groundwater Rn flux from the Rn mass balance in Lake 850 in September ($178 \pm 39\,\mathrm{Bq\,m^{-2}\,d^{-1}}$) by the average groundwater Rn flux obtained from the lake survey in September ($74.5 \pm 55\,\mathrm{Bq\,m^{-2}\,d^{-1}}$, *C. Olid, unpublished data*). The derived ratio (2.39) was then multiplied by the mean groundwater Rn

fluxes from the lake survey to model the groundwater Rn fluxes in Lake 850 for those months were Rn measurements were not available (June, July and September). Rn fluxes through groundwater during the ice-covered period were assumed to be 40% lower than those measured in September (*C. Olid, unpublished data*; Table A1). The groundwater Rn flux from November to April was assumed to be constant and equal to April estimations.

**Table A1.** Estimated Rn fluxes in August and September with the derived water discharges through groundwater based on the Rn mass balance.

| Measure data | $\mathrm{Bq\,m^{-2}\,d^{-1}}$ | SD | Month | $\frac{Rn_{L850}}{Rn_{10lakes}}$ | Q [$\mathrm{m^3\,d^{-1}}$] | SD |
|---|---|---|---|---|---|---|
| 08/2019 | 164 | 51 | 8 | | 309 | 108 |
| 09/2019 | 178 | 39 | 9 | 2.39 | 335 | 92 |

**A2    Groundwater DSi and $\delta^{30}Si$ calculations during the ice-covered period**

The groundwater concentration $c_{gw}$ in during the ice-covered period was calculated from the groundwater discharge, the lake volume from the water balance, and the lake DSi differences between September and March though a mixing model:

$$c_{gw} = \frac{(c_{Mar}(V_{Sept} + V_{gw})) - (c_{Sept} \times V_{Sept})}{V_{gw}} \tag{A1}$$

where $c_{Mar}$ is the lake concentration in March, $c_{Sept}$ is the lake concentration in September, $V_{Sept}$ is the lake volume in September, and $V_{gw}$ in is the total volume of water brought by groundwater in 8 months. The total water volume brought by

groundwater in 8 months was calculated from the modelled groundwater winter discharges (Figure 2). The lake volume in September is taken from the water balance model, where the modelled groundwater discharges were used (Figure 5 and A1, green line).

Similarly, the $c_{gw}$ in during the ice-covered period in the scenario with continuous BSi flux to the sediment for period of 8 month was calculated by adding flux into the sediment into the mixing model:

$$c_{gw} = \frac{(c_{Mar}(V_{Sept} + V_{gw})) - (c_{Sept} \times V_{Sept}) + \phi_{BSi}}{V_{gw}} \tag{A2}$$

where the $\phi_{BSi}$ is the total flux of BSi to sediment in 8 months. The BSi flux into the sediment for 8 months was calculated as a sum of the continuous monthly BSi flux from September until March.





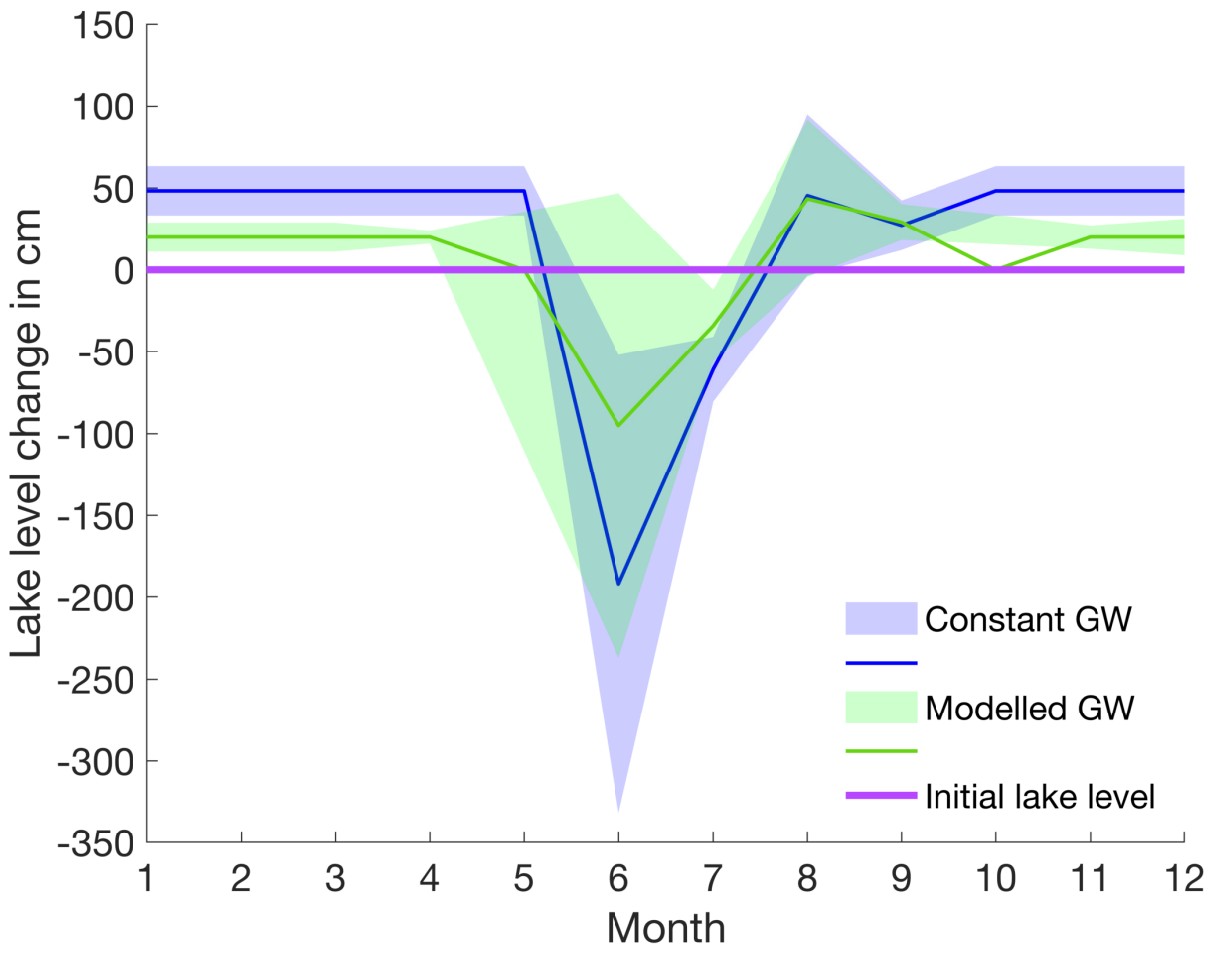

**Figure A1.** Estimated lake-level variation between neighbouring months throughout the year with uncertainties as shading. No lake-level change is depicted by the solid purple line. The blue line presents lake-level increase or decrease from one to another month with constant groundwater discharge (scenario i), and the green line is the rate of lake-level variation with modelled groundwater discharges (scenario ii).

The isotopic composition of the groundwater during the ice-covered period, based on measured data was calculated as:

$$\delta^{30}\mathrm{Si_{gw}} = \frac{(\delta^{30}\mathrm{Si_{Mar}}(c_{Sept} \times V_{Sept})) + (c_{gw} \times V_{gw}) - (c_{Sept} \times V_{Sept} \times \delta^{30}\mathrm{Si_{Sept}})}{c_{gw} \times V_{gw}} \tag{A3}$$

where $\delta^{30}\mathrm{Si_{Mar}}$ is the lake isotopic composition in March, $\delta^{30}\mathrm{Si_{Sept}}$ is the lake isotopic composition in September, $c_{Sept}$ is the lake concentration in September, $c_{gw}$ in is the concentration of groundwater during the ice-covered period (eq. A1 or A2, depending on model), $V_{gw}$ in is the total volume of water brought by groundwater in 8 months, and $V_{Sept}$ is the lake volume in September.



## A3   Silicon isotope mass balance – $\delta^{30}Si_{gw}$ calculation

Due to the high groundwater input in Lake 850 proven by the Rn mass balance ( see section Results: 5.2 Groundwater discharge), the inlet $\delta^{30}$Si does not represent the initial $\delta^{30}$Si used by diatoms. Therefore, the initial $\delta^{30}$Si of DSi is a mixture of groundwater $\delta^{30}$Si and inlet $\delta^{30}$Si flux weighted. The $\delta^{30}$Si$_{\text{initial}}$ in was calculated from $\delta^{30}$Si$_{\text{postuptake}}$, which equals to $\delta^{30}$Si$_{\text{lake}}$ as:

$$\delta^{30}\text{Si}_{\text{initial}} = \delta^{30}\text{Si}_{\text{postuptake}} + {}^{30}\varepsilon \times (1 - \frac{\text{c}_{\text{out}}}{\text{c}_{\text{initial}}}) \tag{A4}$$

Further the groundwater $\delta^{30}$Si$_{\text{gw}}$ which fits the measured data and keeps the steady-state, was calculated as:

$$\delta^{30}\text{Si}_{\text{gw}} = \frac{\delta^{30}\text{Si}_{\text{initial}} \times ((\text{c}_{\text{in}} \times \text{Q}_{\text{in}}) + (\text{c}_{\text{gw}} \times \text{Q}_{\text{gw}})) - (\text{c}_{\text{in}} \times \text{Q}_{\text{in}} \times \delta^{30}\text{Si}_{\text{in}})}{\text{c}_{\text{gw}} \times \text{Q}_{\text{gw}}} \tag{A5}$$

## Appendix B:  Mass balance models: Extreme Si and Si isotope mass balances

The Si and Si isotopic mass balances models were tested for two extreme scenarios to model the highest and the lowest possible concentration of groundwater brought into the lake. Further, a scenario based on recent diatom growth season is modelled (Table

B1). The DSi concentration and isotopic composition from the inlet and outlet streams are similar in all three scenarios. The groundwater DSi concentrations and isotopic composition are calculated from the groundwater fluxes influenced by the three potential BSi fluxes into the sediment, representing three possible lengths of diatom production. All scenarios are using the open system isotopic model (Varela et al., 2004) to describe the effect of diatom production on the lake water $\delta^{30}$Si signature. The difference between the first and second scenario is the BSi flux into the sediment: (1) considers BSi flux into the sediment

throughout the whole year representing lack of ice-covered period, and (2) BSi flux into sediment is present only from June until September (Shemesh et al., 2001). Scenario (3) utilizes the open system isotopic model only for June, with no diatom production the rest of the year, and thus no fractionation in the lake, which describes lake behaviour with only short ice-free period. Here we describe only scenario 1 and 3, whereas in the main text scenario 2 is presented and discussed.

**Table B1.** A summary of all 3 scenarios, which were examined through Si and Si isotope mass balance models

| scenario | BSi flux time | daily BSi flux | range $c_{GWin}$ | range $\delta^{30}Si_{GWin}$ | DSi% consumed by production | range $\delta^{30}Si_{BSi}$ |
|---|---|---|---|---|---|---|
| 1 | 12 months | $0.58\,\text{kg day}^{-1}$ | 1.40 to $11.29\,\text{mg l}^{-1}$ | $-0.63$ to $1.38‰$ | $39-57\%$ | $-0.49$ to $-0.01‰$ |
| 2 | 4 month | $1.77\,\text{kg day}^{-1}$ | 3.51 to $6.95\,\text{mg l}^{-1}$ | $-0.55$ to $1.45‰$ | $63-79\%$ | $-0.49$ to $-0.01‰$ |
| 3 | 1 month | $7.08\,\text{kg day}^{-1}$ | 0.37 to $8.99\,\text{mg l}^{-1}$ | $-0.04$ to $1.45‰$ | $0-84\%$ | $-0.21‰$ |





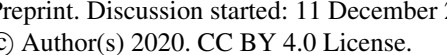

**Figure B1.** Inlet (blue), outlet (red), groundwater (cyan), and the BSi (magenta) fluxes, concentrations and isotopic composition. Scenario 1: A-C The constant BSi flux into sediment (A) influences the groundwater DSi concentration (B) and the silicon isotopic composition (C). Scenario 2: D-F present the effect of BSi flux adjusted to the diatom bloom season of 4 months (D) on the groundwater concentration (E) and isotopic composition (F). Scenario 3: G-I The diatom bloom represented by the BSi flux into the sediment is restricted to 1 month only. During June all BSi accumulated for 1 year is produced and the groundwater concentration (H) and the isotopic composition (I) is affected.





## B1 Scenario 1: 12 months of BSi flux into sediment

A scenario assuming a constant BSi flux to the sediment throughout the year (magenta line, Figure B1A) simulates a situation when climate is warming, and the diatom growth season is prolonged to maximum. Additionally, this scenario was characterized by the minimal groundwater fluxes and DSi concentrations. The DSi removal by diatoms is of $1.21 \pm 0.62 \, \mathrm{mg\,l^{-1}}$ $SiO_2$ monthly (magenta points, Figure B1B). Therefore, with an added BSi flux of $0.58 \pm 0.29 \, \mathrm{kg}$ $SiO_2$ per day, the lake inlet does not supply sufficient DSi for diatoms to grow. The groundwater DSi concentration is calculated as the DSi flux needed to keep

the lake balanced and sustain the diatom production. The groundwater flux of DSi varies from $0.43 \pm 0.51$ to $2.20 \pm 1.35 \, \mathrm{kg}$ $SiO_2$ per day, depending on the season (cyan line, Figure B1A). The highest groundwater DSi flux occurs in June, followed by a decreasing trend towards August, when it reaches the minimum. From August until November, the groundwater DSi flux increases and is stabilized after November, and it is constant until May. From the calculated groundwater flux, the groundwater concentration is between $1.40 \pm 1.59 \, \mathrm{mg\,l^{-1}}$ to $2.31 \pm 1.03 \, \mathrm{mg\,l^{-1}}$ during the ice-free period, and, when combined with the

lake DSi deficiency at the end of the season, it is $11.29 \pm 1.07 \, \mathrm{mg\,l^{-1}}$ (cyan line, Figure B1B) in the ice-covered period.

In scenario 1, with constant BSi flux into the sediment during the whole year of $0.58 \pm 0.29 \, \mathrm{kg}$ $SiO_2$ per day, high superficial and groundwater discharges occur in June, with DSi concentrations of 2.34 and $2.28 \, \mathrm{mg\,l^{-1}}$, respectively (Figure B1B). The stream inlet has a light isotopic signature of $\delta^{30}Si_{in} = 0.02 \pm 0.10‰$. The initial DSi available for diatoms is a mixture of the groundwater and the stream inlet, with $\delta^{30}Si_{inital} = 0.36 \pm 0.29‰$. The groundwater was calculated from the $\delta^{30}Si_{inital}$

to have an isotopic signature of $\delta^{30}Si_{gw} = 0.46 \pm 0.51‰$ in June. Thus, the expected BSi isotopic signature was calculated to be $-0.21 \pm 0.41‰$, which is within the range of average measured $\delta^{30}Si_{BSi} = 0.07 \pm 0.43‰$ in the top sediment layers. The diatom production consumes approximately 48% of the DSi influx in June.

Although groundwater discharge culminates in July, compared with the decreasing trend in the stream inlet, the isotopic composition of the lake in July is influenced by both the groundwater and the stream. The DSi concentration of the inlet

is $4.79 \pm 0.03 \, \mathrm{mg\,l^{-1}}$, but with 4 times lower discharge than groundwater. The calculated groundwater DSi concentration from the steady-state model is only $1.54 \pm 1.39 \, \mathrm{mg\,l^{-1}}$ (Figure B1B). Further, the initial isotopic mixture for diatom growth $\delta^{30}Si_{initial} = -0.02 \pm 0.10‰$ is composed of the stream $\delta^{30}Si_{in} = 0.72 \pm 0.10‰$ and the groundwater $\delta^{30}Si_{gw} = -0.63 \pm 0.75‰$ (Figure B1C). The expected BSi isotopic signature is $\delta^{30}Si_{BSi} = -0.49 \pm 0.41‰$, which still falls within the average measured $\delta^{30}Si_{BSi} = 0.07 \pm 0.43‰$ in the sediment. The diatom production in July consumes 57% of the DSi.

In August, the isotopic composition of the stream inlet, lake, and outlet are similar. The DSi concentration of the inlet is at its maximum, with an isotopic composition of $0.78 \pm 0.15‰$, but due to a very low inlet discharge it is not affecting the lake. The concentration and isotopic signature of the outlet and the lake are almost identical, thus, the groundwater input is $1.40 \pm 1.59 \, \mathrm{mg\,l^{-1}}$ of DSi (Figure B1B), with an isotopic signature of $\delta^{30}Si_{gw} = 0.14 \pm 0.73‰$ (Figure B1C). The expected BSi isotopic signature is $\delta^{30}Si_{BSi} = -0.31 \pm 0.67‰$, which is in agreement with the average measured $\delta^{30}Si_{BSi}$ in the diatoms

from sediment. The diatom production in August consumes 39% of the lake DSi.

September is the last month before the lake is ice-covered. There is no stream inlet, as the watershed is snow-covered. The groundwater input, with a concentration of $2.31 \pm 1.03 \, \mathrm{mg\,l^{-1}}$, is four times higher than the removal by the lake outlet.





This suggests that the lake-level is changing throughout seasons, which is not considered in any of the Si mass balance and isotopic models examined. The lake DSi of $1.37 \pm 0.04 \, \mathrm{mg\,l^{-1}}$ is fully influenced by groundwater and diatom production. The groundwater isotopic signature is $0.65 \pm 0.55‰$, and the diatom production is using $41\%$ of the lake DSi. The expected BSi isotopic signature is $\delta^{30}\mathrm{Si_{BSi}} = -0.01 \pm 0.47‰$, which is in agreement with the average measured $\delta^{30}\mathrm{Si_{BSi}}$ in the diatoms from sediment.

This scenario assumes that the groundwater concentration during the ice-covered lake is recharging the lake DSi, while the BSi flux into the sediment is still present (Figure B1B). Applying the mixing model (equation A2 and A3), groundwater DSi concentration ($11.29 \pm 1.07 \, \mathrm{mg\,l^{-1}}$), groundwater discharge, lake volume change during the ice-covered period, and the difference of the isotopic composition of the lake water between September ($1.02 \pm 0.24‰$) and March ($1.27 \pm 0.10‰$), the isotopic signature of the groundwater is calculated to be $1.45 \pm 2.58‰$ (Figure B1C).

## B2    Scenario 3: only 1 month of BSi flux into the sediment

The third scenario is based on the inlet and outlet DSi fluxes but assumes that diatom production occurs only in June. This scenario could occur if the climate would experience cooling and the diatom growth period would be extremely shortened. Additionally, this scenario demonstrated the highest groundwater concentrations during the growing season. The rest of the year diatom production, and so the BSi flux into the sediment, is negligible or zero. Therefore, the yearly accumulated BSi settles into the sediment within one month, which yields a BSi flux of $7.08 \pm 3.62 \, \mathrm{kg \, SiO_2}$ per day (magenta line, Figure B1G). In this scenario, groundwater input must be from $0.15 \pm 0.37 \, \mathrm{kg \, SiO_2}$ to $8.70 \pm 4.59 \, \mathrm{kg \, SiO_2}$ per day, and the DSi concentration ranges between $0.37 \pm 0.69 \, \mathrm{mg\,l^{-1}}$ to $8.99 \pm 4.35 \, \mathrm{mg\,l^{-1}}$ during the ice-free period (cyan line, Figure B1H). Similar to the second scenario (in the main text), to restore the lake DSi concentration during the ice-covered period from lake-October to mid-June, groundwater DSi concentration is around $6.95 \pm 4.90 \, \mathrm{mg\,l^{-1}}$.

Scenario 3 assumes the BSi flux into the sediment occurs only in June, and the rest of the year there are no processes causing stable Si isotope fractionation. This scenario originates from data in August and September, when the $\delta^{30}\mathrm{Si}$ of inlet, outlet and the lake are very similar. Only in June is there fractionation between the lake stream inlets and the lake, which is described by the open-system-fractionation model. Therefore, the groundwater concentration in June increases to $8.99 \pm 4.35 \, \mathrm{mg\,l^{-1}}$ (Figure B1H), with an isotopic signature of $-0.04 \pm 0.52‰$ (Figure B1I) to sustain the diatom production represented by BSi flux into the sediment. The production consumes $84\%$ of the available DSi.

In July, August, and September the groundwater DSi concentration is low, as the lake does not have any production, thus no demand on the DSi. The isotopic composition of the groundwater is $0.23 \pm 1.41‰$, $0.75 \pm 2.83‰$, and $1.02 \pm 0.53‰$, respectively (Figure B1I). High uncertainties in the isotopic composition of the groundwater reflect the uncertainties in the stream and groundwater discharges and fluxes.





## Appendix C: Discussion: Scenarios evaluation

Scenarios 1 and 3 of Si mass balance (Table B1, Figure B1A and G) are demonstrating how the groundwater concentration
would change with changes of length of diatom production. It is likely that the diatom growth season would be driven by
the changes in climate and thus the ice-free period length. Our models aimed to estimate the changes in the lake DSi and Si
balance in those extreme changes of growing season driven by changes in climate. However, the groundwater concentrations
are commonly higher than the superficial streams (Frings et al., 2016; Maavara et al., 2018; Opfergelt et al., 2011), which is
not the case in scenario 1 and 3. The groundwater DSi concentrations are lower than in the stream inlet during the ice-free
period in those two scenarios (Figure B1B and B1H), which suggest that those scenarios have either missing or surplus data
of the inlet and outlet DSi concentration and discharges. A more complex model with variable discharges of groundwater and
stream inlets and outlet depending on precipitation end evaporation changes would be needed. Therefore, those two scenarios
bring only a rough estimate hinting the changes in DSi and Si isotopic mass balances connected to changes in climate.