# Peer review of "Modern silicon dynamics of a small high-latitude subarctic lake"

_Biogeosciences, 2020_

## Referee Comment (RC1) · Anonymous Referee #1 · 30 Dec 2020

The manuscript by Zahajská et al provides insights into the silicon dynamics of a small subarctic lake in Sweden. Through this, the authors demonstrate the considerable importance of groundwater in lake silicon cycling in order to account for the changes in their monitoring record.

Whilst the amount of raw data in the paper is relatively low, the findings will be of considerable interest to biogeochemists in future silicon cycling studies (both those working on both large/small as well as high/low latitude lakes). Overall, I'm keen to see this novel study published, however there are some issues (most of them minor) which the authors would benefit from considering:

* Line 57-59: I assume you mean "surface atmospheric" temperature.

[Figure]

\* Line 58: growing season - I assume you mean aquatic growing season.

\* Line 118: The assumption of a steady state needs to be explained/justified better

\* I would suggest moving Section 4 before Section 3, but this is up to the authors to decide.

\* Line 314: "average" - replace with "mean"?

\* Line 314: I don't see the need for this sentence - writing about mean sedimentation rates in the core seems unnecessary given the following lines which look at temporal variability in the core.

\* Line 317-318: I'm confused here: 1) According to the text in these lines the number of analysed samples is 3 ("n = 3"), but at line 231 n = 25; 2) What ages/depths are the diatom silicon isotopes samples? This data should be plotted in Figure 4. 3) Why is a mean of all diatom silicon isotope samples used in subsequent calculations (see comment further below)? Why not do the mass balance calculations on each sample individually? Doing it individually on the uppermost (core top sample) would be particularly good in providing a value that is more analogous to the modern data used in the rest of the paper.

\* Section 6.1 - are modelled seasonal/annual lake level changes feasible and/or supported by observations. Given a modern lake depth of 8 m (line 61), some of these lake level changes seem (to me) fairly extreme.

\* Line 337 - this sentences seems very simplistic and would benefit from being explored/interrogated further.

\* Line 341-347: some repetition exists within this section of text. \* Line 354: "ans" = "and"?

\* Line 358-366: What happens if you do this for each sediment depth you have a BSI and diatom d30Si sample for? Or what happens if you do this just for the youngest

(core top sample)? Would this be better for examining modern silicon fluxes in the lake rather than using mean values over the top 8 cm which covers the last 150 years?

* Line 458: Change "The yearly BSI flux would increase" to "The yearly BSI flux would need to increase"?

* Figure 1: add the year that samples were collected to the legend in the bottom right of the right panel.

* Figure 2: Consider using different colours to show the modelled QGW (line 296) and the measured QGW in August/September and then change the figure caption accordingly. Initially, the same colours on the plot for QGW confused me.

* Figure 6: Use different colours that make it easier to distinguish between each variable.

[Figure]

---

## Referee Comment (RC2) · Anonymous Referee #2 · 1 Jan 2021

The manuscript by Zahajská et al. presents a interesting point describing the silicon dynamics in a high-latitude lake. The authors quantified the groundwater and river inputs by using Si isotope and 222Rn mass balances through the years. Groundwater contributed 3 times higher than the rivers. The methods could be used for other water bodies. In general, It is a nice piece of work to be published in Biogeosciences. However, the current manuscript version does not reach the level of Biogeosciences. Especially the interpretation of the raw data and the discussion focus too much on the mass balance for the whole lake.

Considering 79% of DSi input is consumed by diatoms. I would like to see more interpretation about the changes of DSi and BSi concentration, for example the DSi concentration decrease in the beginning of the growing season, however the DSi consumed

by diatoms and BSi flux to sediment increased. To my knowledge, the temperature involved the process.

There are a number of other changes and theoretical issues that should be considered. 1. The authors claimed the mechanisms responsible for the diatom-rich sediment formation in the Lake, I only see the results and discussion based on water/silicon isotope/ Radon mass balances. I would like to see more discussion about the water-sediment interaction. How the temperature involved the primary production of diatom and dissolution of detritus, especially during the growing season. 2. Considering the residence time of the lake is around 2 months during the growing seasons and most of the biogeochemical processes happened during the same period, water samples from streams and lakes were collected from June to September 2019, while the Si isotope mass balance based on an age depth model. To me there is a huge uncertainty. 3. I am confused by the DSi consumed by diatoms in Figure 6 B, is it constant? Can you also add the monthly temperature in the figure? 4. Considering the high accumulation of BSi, any idea to include the legacy assessment based on your data set? I did'n see this in the discussion. 5. I am interested in the BSi flux in Figure 4. Can you explain the change for the period 1803-2019? Maybe you can get some clue from DOI: 10.1038/ncomms1128 or mainly because of the sedimentation rate?

Specific comments L 10-11 Switch section 3 And 4. Maybe considering combining to one section. L 58 the temperature in 2018-2019 should be 10.1 L 70 Compress this section by move more details to SI. L244-245 The text read like out of place. L248 Can you add the rough flow velocity of the Lake during the growing season? I am curious the net sedimentation rate of BSi in the lake. More interpretation about how the residence time impact the BSi accumulation. L 254-257 Based on this part, discuss how does BSi and temperature involved in the seasonal change of DSi in the below discussion part. L 478 fine-grain should be fine-grained.

Some references maybe the authors would benefit how to explore the mechanisms especially the water-sediment dynamics. 1. A simplified algorithm for calculating benthic nutrient fluxes in river systems 2. Exploring Long-Term Changes in Silicon Bio-geochemistry Along the River Continuum of the Rhine and Yangtze (Changjiang) 3. Historical land use change has lowered terrestrial silica mobilization

---

## Referee Comment (RC3) · Anonymous Referee #3 · 18 Jan 2021

General comments: This is an interesting manuscript that used multiple lines of evidence, including element, isotope, and water mass balances to inform their interpretation. It will be a valuable contribution to the scientific literature for its insights into both silicon biogeochemistry in these unique high arctic lake environments and the importance of groundwater as a source of dissolved silicon to lakes. I appreciated the scenario analyses that were undertaken for some of the calculations considering the limitations to the dataset.

I believe that the manuscript is of sufficiently high quality in terms of its methods and interpretation of its results and is a novel contribution to Biogeosciences. I have some minor comments for the authors to consider before it is ready for publication in Biogeosciences.

[Figure]

Specific comments:

Should appendices A, B and C be placed in the supplementary materials rather than in the main manuscript text file?

I recommend using the term "isotope ratio" instead of "isotopic signature" throughout the manuscript. I refer to Waterbirds, 35(2):324-331 (2012). https://doi.org/10.1675/063.035.0213 for isotope ratio reporting guidelines, which summarizes the recommendations of the Commission on Isotopic Abundances and Atomic Weights (CIAAW) of the International Union of Pure and Applied Chemistry (IUPAC).

Lines 3-4: Can you be more specific instead of just saying "factors and mechanisms". I realize that this is just the abstract.

Line 4: I somewhat disagree with the statement "While BSi formation and preservation is expected to occur in silica rich environments with high dissolved silicon (DSi) concentrations such as volcanic and hydrothermal inputs, the factors and mechanisms explaining high DSi and BSi concentrations in lakes remain unclear." What qualifies as a high DSi concentration? DSi concentrations are relatively high (in the 100 $\mu$M, or 6 mg SiO2 l-1, range) in most groundwater, rivers, and lakes that drain catchments with siliceous bedrock and soils, but also in most catchments overall. BSi formation and preservation also happens in these environments. I understand that this is just the abstract and you qualify these statements more in the introduction. Obviously, as you point out and discuss, the BSi concentrations in the lake's sediments are higher than many other lakes because of the low sedimentation rate, which is unique to high arctic lakes. But I don't think that you can say that the high DSi and BSi concentrations you observed in the lake water column are abnormally high compared to a temperate or tropical freshwater lake.

Even in your short summary on Biogeosciences, you say "The processes responsible for large accumulation of siliceous shells of a single-cell algae – diatoms – in lake deposits are poorly understood." I would somewhat disagree with this statement as

well. BSi preservation occurs when BSi production and deposition is greater than its dissolution. To me, it seems that the more specific description of the research gap is that "the Si budget of this the lake had not been fully characterized before to establish the drivers of BSi accumulation in this environment, which is a more specific description of what your research objectives were rather than just saying "that the processes responsible for BSi accumulation in sediments are poorly understood"

Line 30: What physical processes are you referring to that remove DSi from solution? Can you be more specific?

Line 117: Regarding "and 1000 is unit conversion.": Please specify what the units of this unit conversion factor are.

Line 119: When you say "Assuming steady-state ($\Delta$DSi = 0)": Can you elaborate more and be more specific for readers not familiar with steady state conditions what you are talking about. Something to the effect of "Assuming that the DSi fluxes in the lake are at steady state (i.e., the sum of the input fluxes is equal to the sum of the output fluxes and therefore that $\Delta$DSi = 0)"

Lines 181-186: What method did you use to measure the DSi content in your alkaline extractions? I assume that it was by the automated molybdate-blue method that was used to measure the DSi content in you water samples, but please add this detail to clarify for readers. Please also give a bit more detail about the extraction method and its requirement for a mineral correction factor. You could even do this by saying "Si-containing minerals" on line 184 instead of just "minerals"

Lines 403-408: Could these lines go in the Model uncertainties discussion section given that you have a section dedicated to discussing these uncertainties?

Lines 471-480: I think this discussion, which is the conclusions, belongs in its own discussion section about comparing your lake's environmental controls on BSi accumulation with that of other lakes and environments where high BSi accumulation has

been observed, and the implications of this finding (regarding the fact that groundwater is likely a significant source of DSi to lakes and should be considered in lake Si budgets). Then, one sentence in your conclusions can summarize this discussion section.

Lines 474-476: Reconsider your discussion about high autochthonous carbon production in arctic lakes here. Autochthonous carbon production rates (that is, carbon fixation rates) are not independent of diatom DSi uptake rates and BSi production and burial rates.

A related thought, which might be outside of the scope of this work, but I thought I'd share nonetheless: Have you thought about the implications of your work for our understanding of carbon burial in arctic lakes. Specifically, I am thinking – could the high diatom productivity, driven by high DSi supply via groundwater, drive increased carbon burial in lakes with high groundwater discharge relative to lakes with less groundwater discharge (and therefore less DSi supplied)? Could this be a research hypothesis that you could put forth in this paper for others to answer? In other words, could you suggest that the implications of your work extend beyond understanding Si biogeochemistry and can inform our understanding of carbon fixation and burial rates in arctic lakes? It is especially interesting that you have sediment TOC concentration data, and they appear to be correlated with the sediment BSi concentrations.

You could refer to these papers: Wang, B., Liu, CQ., Maberly, S. et al. Coupling of carbon and silicon geochemical cycles in rivers and lakes. Sci Rep 6, 35832 (2016). https://doi.org/10.1038/srep35832 Krause, J.W., Schulz, I.K., Rowe, K.A. et al. Silicic acid limitation drives bloom termination and potential carbon sequestration in an Arctic bloom. Sci Rep 9, 8149 (2019). https://doi.org/10.1038/s41598-019-44587-4

Technical corrections:

A general technical comment: Check your pluralization of words and whether you need an article (a/an or the) in front of your nouns. For example, I would recommend that stream discharge does not need to be pluralized. I made note of some of these grammar/technical corrections, but not of all of them.

Some suggestions for modifications to terminology used or sentence phrasing:

Line 1: "concentrations occur" instead of "concentration occurs"

Line 6: "stream discharge" instead of "stream discharges"

Lines 8-9: I would recommend re-phrasing to clarify that one fifth of the DSi that would otherwise be exported to the ocean is retained by lakes, something like: "estimated to retain one fifth of the annual DSi terrestrial weathering flux that would otherwise be delivered to the ocean" instead of "estimated to retain one fifth of the annual DSi delivery into the ocean"

Line 9: "DSi inputs being 3 times higher" instead of "DSi inputs 3 times higher"

Line 17: "dissolved silicic acid, H4SiO4, expressed here as dissolved silicon (DSi), and . . ." instead of "dissolved silicic acid H4SiO4, expresses here as dissolved silicon (DSi), and . . ."

Line 23: "One example is high-elevation" instead of "One example is the high-elevation"

Line 24: "high BSi concentrations" instead of "large BSi concentrations"

Line 30: "removed" instead of "taken up" (given that you refer to both physical and biological processes)

Lines 45-46: "Therefore, stable Si isotopes are an" instead of "Therefore, the stable Si isotopes provide an"

Line 60: I think you mean "850 m above the tree-limit" instead of "850 m a.s.l., above the tree-limit"

Line 184: Change wording: "As there were no changes in the amount of total Si extracted during the time course of dissolution" instead of "As no changes in the amount of total Si extracted during the time course of the dissolution"

Line 301: "accumulation" instead of "accumualtion"

Line 363: "The production consumes 63%" instead of ": "The production consumes from 63%"

Line 381: "The significance of groundwater-sourced DSi to the lake's Si cycle" or "The significance of groundwater-sourced DSi to Lake 850's Si cycle" instead of "The significance of groundwater on lake Si cycle"

Lines 410-435: Please specify every time that you mention the modelled groundwater $\delta$30Si values that they are modelled. For example: please add this specification to lines 416 and 417.

Figures:

You use the shortform HTH core in the caption for Figures 3 and 4 but you don't explain this shortform in the text. Please change this.

Can you increase the font size for the axis labels and numbers in Figure 3? And could the inset figures go in the Supplementary Information? They seem extraneous and are not thoroughly explained or discussed in the main text. I understand if this is difficult to do because of the default graphical output of the PLUM package.

Can you increase the resolution of Figure 5? It seems to be a lower resolution compared to the other figures, and the x-axis numbers are somehwat cut off at the top.

---

## Author Comment (AC1) · 12 Feb 2021

We thank Referee 1 for suggested edits and clarifications. We followed these suggestions carefully for the revision. We enclose a full account of our responses (normal font) to the comments and suggestions of the reviewer (in italics), including references to all changes in the manuscript.

*The manuscript by Zahajská et al provides insights into the silicon dynamics of a small subarctic lake in Sweden. Through this, the authors demonstrate the considerable importance of groundwater in lake silicon cycling in order to account for the changes in their monitoring record. Whilst the amount of raw data in the paper is relatively low, the findings will be of considerable interest to biogeochemists in future silicon cycling*

*studies (both those working on both large/small as well as high/low latitude lakes). Overall, I'm keen to see this novel study published, however there are some issues (most of them minor) which the authors would benefit from considering:*

*Line 57-59: I assume you mean "surface atmospheric" temperature*
The text has been corrected according to the reviewer's comment (Line 56-57).

*Line 58: growing season - I assume you mean aquatic growing season.*
The text has been corrected according to the reviewer's comment (Line 57).

*Line 118: The assumption of a steady state needs to be explained/justified better.*
The assumption of steady-state was specified and explained as following: Assuming that the lake is in steady-state, which means that sum of input DSi fluxes equals to sum of output Si fluxes, thus $\Delta_{DSi} = 0$, DSi concentration in groundwater was then calculated by dividing . . . (L125-126)

*I would suggest moving Section 4 before Section 3, but this is up to the authors to decide.*
We understand the reviewer's suggestion. However, we believe that having a theoretical section "Numerical analyses" before Material and Methods helps the reader to understand specific types of sampling and analyses. For this reason, we believe that the structure of the manuscript is logic as it is now and we didn't take any action here. However, we will agree on moving these sections if the editor considers it necessary.

*Line 314: "average" - replace with "mean"?*
The entire manuscript text has been corrected according to the reviewer's comment. All "average" were replaced by "mean"

*Line 314: I don't see the need for this sentence - writing about mean sedimentation rates in the core seems unnecessary given the following lines which look at temporal variability in the core.*
The sentence now reads "From MAR and BSi wt% we estimated the BSi accumulation

rate ($\phi_{BSi}$)" (Line 325). We have also added one sentence below, explaining why we do calculate mean BSi accumulation rates from mean MAR and BSi wt%: "The mean BSi accumulation rate for the entire gravity core of $2.9 \pm 1.5$ mg $_2.cm^{-2}.yr^{-1}$ was used as the BSi flux to sediment in the mass balance models." (Line 327-328)

*Line 317-318: I'm confused here:*

1. *According to the text in these lines the number of analysed samples is 3 ("n = 3"), but at line 231 n = 25;*
   We have now specified that the mean diatom isotopic signature is calculated from 3 measured samples: n=26 (updated) refers to all measured samples, which includes water and diatoms. We have clarified this by stating that "Each sample was measured three times, bracketed by NBS-28 in between, and full chemical replicates for diatom (n=3) and water samples (n=23) were measured for 65% of all samples (total measurements n=137)" (Line 240-241)

2. *What ages/depths are the diatom silicon isotopes samples? This data should be plotted in Figure 4.*
   We would like to clarify here that the diatom isotope data originate from another core. That is why these data are not plotted in Figure 4, which presents only data from the gravity core. Adding the 3 isotopic data points from this other core in Figure 4 can be done only based on their ages, which is highly dependent on both age-depth models. To clarify this, we have added a sentence (L 219-220) in the Methods section: Clean diatom material (n=3) from a published core taken in 1999 (Shemesh et al., 2001) was used to determine the stable silicon isotope ratio in sedimentary diatoms and then used in mass balance models.

3. *Why is a mean of all diatom silicon isotope samples used in subsequent calculations (see comment further below)? Why not do the mass balance calculations on each sample individually? Doing it individually on the uppermost (core top sample) would be particularly good in providing a value that is more analogous*

*to the modern data used in the rest of the paper.*

We use mean $\delta^{30}Si$, because the diatom material originates from another core, which is dated by the $^{14}C$ method, thus the topmost ages of the core are only extrapolated from the age-depth model. Rather than trying to align our $^{210}Pb$ dated gravity core with the $^{14}C$ dated piston core, we use the mean. We have no $\delta30Si$ diatom data from the gravity core nor 1-cm resolution in the other core from 1999.

4. *Section 6.1 - are modelled seasonal/annual lake level changes feasible and/or supported by observations. Given a modern lake depth of 8 m (line 61), some of these lake level changes seem (to me) fairly extreme.*

The modelled lake-level changes have high uncertainties as shown in Figure 5 and A1, thus extreme lake-level changes are unlikely to happen, but based on our data, we still do not exclude them.

*Line 337 - this sentence seems very simplistic and would benefit from being explored/ interrogated further.*

This sentence is meant to introduce the assumption for the mass balance model. More details about all processes responsible for the BSi accumulation are discussed in the Discussion section 7.2. We have modified the text according to the reviewer's comment. Now the text reads as: Based on the steady-state assumption, the BSi accumulation occurs in conditions when the total DSi influx is higher that the stream DSi outflux (Lines 349-350).

*Line 341-347: some repetition exists within this section of text.*

Unfortunately, we have not found repetition in this section of text. We are willing to modify the text if concrete suggestions of repetition is suggested (now Lines 354-360).

*Line 354: "ans" = "and"?*

The text has been modified and corrected according to the reviewer's comment (Line 367).

*Line 358-366: What happens if you do this for each sediment depth you have a BSI and diatom d30Si sample for? Or what happens if you do this just for the youngest (core top sample)? Would this be better for examining modern silicon fluxes in the lake rather than using mean values over the top 8 cm which covers the last 150 years?*

Although the suggestion by the reviewer is interesting, we are not able to do this as we do not have 1-cm resolution of $\delta^{30}Si$. The alignment of the piston and gravity core could be done by comparing the age-depth relationship, however, this is far from being an accurate method as each core has been dated using a different method. This alignment, then, would not increase the resolution, and rather increase the uncertainties for the record, especially for the most recent (the last 150 years) layers. Therefore, we "homogenize" the $\delta^{30}Si$ data for last 150 years.

*Line 458: Change "The yearly BSI flux would increase" to "The yearly BSI flux would need to increase"?*

The text has been corrected according to the reviewer's comment (Line 482).

*Figure 1: add the year that samples were collected to the legend in the bottom right of the right panel.*

We have added the years of sample collection into the legend of Figure 1.

*Figure 2: Consider using different colours to show the modelled QGW (line 296) and the measured QGW in August/September and then change the figure caption accordingly. Initially, the same colours on the plot for QGW confused me.*

We have changed the color of measured Qgw (black cross now) for better readability as well as readability by color-blind individuals and grayscale print.

*Figure 6: Use different colours that make it easier to distinguish between each variable.*

We have changed the color and symbol (black cross now) of the lake DSi and lake $\delta^{30}Si$ for better readability.

---

## Author Comment (AC2) · 12 Feb 2021

We thank Referee 2 for constructive comments on our manuscript. We followed these suggestions carefully for the revision. We enclose a full account of our responses (normal font) to the comments and suggestions of the reviewer (*in italics*), including references to all changes in the manuscript.

*The manuscript by Zahajská et al. presents a interesting point describing the silicon dynamics in a high-latitude lake. The authors quantified the groundwater and river inputs by using Si isotope and 222Rn mass balances through the years. Groundwater contributed 3 times higher than the rivers. The methods could be used for other water bodies. In general, It is a nice piece of work to be published in Biogeosciences.*

*However, the current manuscript version does not reach the level of Biogeosciences. Especially the interpretation of the raw data and the discussion focus too much on the mass balance for the whole lake.*

We would like to emphasize that the mass balance is the main principle of the manuscript. Therefore, we have clarified the aim of the study in the introduction L 47: "Here, we investigate the diatom-rich sediment formation in Lake 850 through water and silicon mass balances." Thus, the study does not aim to constrain the seasonal diatom production, the factors driving the seasonal changes in diatom production or the in-lake processes. The study focuses on the impact of changes in external sources of Si on the lake Si cycling.

*Considering 79% of DSi input is consumed by diatoms. I would like to see more interpretation about the changes of DSi and BSi concentration, for example the DSi concentration decrease in the beginning of the growing season, however the DSi consumed by diatoms and BSi flux to sediment increased.*

The changes in DSi consumed by diatoms and BSi flux to sediment in the beginning of the growing season are modelled based on measured lake DSi and based on a dated sediment core, which do not have the temporal resolution required for seasonal studies. In addition, we do not have data from sediment traps or any other measurements of seasonal diatom production.

*To my knowledge, the temperature involved the process.*

Although we do not have water temperature data for the entire year, lake water temperature is expected to be stable over the year, as suggested by the air temperature stability shown by Bigler and Hall, 2003 (DOI:10.1016/S0031-0182(02)00638-7). For this reason, we do not think that annual changes in water temperature are a main factor driving the diatom production and accumulation in Lake 850 and we have not included this in the discussion.

We point out in Lines 485-486 that further studies are needed for the entire biogeochemical Si cycle in high-latitude lakes. Further, these studies would require additional

data collection and systematic monitoring to understand the inter-annual factors on BSi accumulation.

*Suggested theoretical issues:*

1. *The authors claimed the mechanisms responsible for the diatom-rich sediment formation in the lake, I only see the results and discussion based on water/silicon isotope/ Radon mass balances. I would like to see more discussion about the water-sediment interaction. How the temperature involved the primary production of diatom and dissolution of detritus, especially during the growing season.*
The objective of this paper was to investigate the diatom-rich sediment formation in Lake 850 through water and silicon mass balances (Line 47). For that reason, water-sediment interactions are out of the scope of this paper. This type of study would require annual lake monitoring using sediment traps, primary production measurements and Si utilization measurements, which has not been done here. Thus, our data do not allow us to conclude more about the sediment development than the information provided by the mass balances and age-depth model from the sediment core.

2. *Considering the residence time of the lake is around 2 months during the growing seasons and most of the biogeochemical processes happened during the same period, water samples from streams and lakes were collected from June to September 2019, while the Si isotope mass balance based on an age depth model. To me there is a huge uncertainty.*
We did construct a model on a monthly basis, however the study aimed to investigate the silicon dynamics on an annual basis. The monthly resolution was chosen to obtain basic knowledge of the system's functioning, rather than to explain the lake processes on a monthly basis. We do use the monthly resolution to constrain all possible factors throughout the year, which may be further applicable on longer-timescales with lower resolution.

3. *I am confused by the DSi consumed by diatoms in Figure 6 B, is it constant? Can you also add the monthly temperature in the figure?*

The DSi consumed by diatoms is calculated from the BSi flux data, which have a resolution of $12 \pm 6$ years. Thus, the resolution is not enough to specify fluxes on a monthly basis. For this reason, we constructed 3 scenarios with 3 different lengths of time of BSi flux to the sediment, which influences the DSi consumed by diatoms. However, those are only 3 scenarios among many, and the lake system is more complex and variable between years. As our study does not aim to describe the processes on a high-resolution scale, such as intra-annual changes, we do not think that including a monthly temperature figure is needed. We focus on constraining the more general functioning of the lake Si cycle based on one year of data.

4. *Considering the high accumulation of BSi, any idea to include the legacy assessment based on your data set? I did'n see this in the discussion.*

Unfortunately, we do not understand the question. We are willing to provide an answer if the question is clarified.

5. *I am interested in the BSi flux in Figure 4. Can you explain the change for the period 1803-2019? Maybe you can get some clue from DOI:10.1038/ncomms1128 or mainly because of the sedimentation rate?*

The relative changes in MAR are more pronounced than changes in BSi, thus the BSi flux is strongly influenced by the MAR. For this reason, we do use mean SR, MAR and BSi wt% in our models. As this is a pristine lake, we do not expect any watershed process to influence BSi accumulation, only the MAR. There is no agriculture in the watershed nor anthropogenic disturbance, hence we do not consider the Nature Communications paper to be relevant here.

*Specific comments:*

*L 10-11 Switch section 3 And 4. Maybe considering combining to one section.*

Sections 3 was kept as the theoretical approach, which aims to introduce the

reasoning for why different samplings were conducted. The sampling schemes are then described in section 4.

*L 58 the temperature in 2018-2019 should be 10.1*
The temperature has been corrected (L 58).

*L 70 Compress this section by move more details to SI.*
We have modified the text according to the reviewer's comment. This section has been removed.

*L244-245 The text read like out of place.*
Part of the sentence was moved to the Study Area section in Lines 59-60, and the text in former Lines 244-245 was adjusted accordingly to: During the ice-free period direct surface precipitation results in $0.65 l.s^{-1}$, which represents only 1.4% of the lake volume, and similar or higher discharges are observed in the stream inlets from July to August (new Lines 253-254).

*L248 Can you add the rough flow velocity of the Lake during the growing season? I am curious the net sedimentation rate of BSi in the lake. More interpretation about how the residence time impact the BSi accumulation.*
Unfortunately, we do not understand what flow velocities the reviewer is interested in. We do not have measurements of flow velocities in the lake. We do not have net sedimentation rates of BSi, as the study does not focus on diatom seasonal production and sedimentation. We use the net accumulation rate of BSi, which is based on the sedimentary data.

The relative retention of DSi as BSi in lakes was shown to be related to the residence time and weakly correlated with the ratio of lake to catchment area (Frings et al., 2014, DOI: 10.1007/s10533-013-9944-z). Results of our model show that Lake 850 accumulates above the expectation, based on data from Frings et al. 2014.

We have added the following sentence into the discussion: Based on the positive

correlation between water residence time and the relative retention of DSi in lakes (Frings et al., 2014), Lake 850 with its DSi retention of $35 \pm 17\%$ of the total DSi inlet and residence time ranging from 0.15 to 1 year, accumulates more DSi as BSi than expected. (Lines 384-386)

*L254-257 Based on this part, discuss how does BSi and temperature involved in the seasonal change of DSi in the below discussion part.*
Unfortunately, we do not have the water temperature data for each season. Moreover, based on the air temperature stability shown by Bigler and Hall, 2003, we expect the lake water temperature to be stable over the year. Further, we look only on sedimentary BSi, thus we cannot assess the effect of temperature on diatom production. Thus, this question is out of scope the study's objectives as we defined it in the Introduction section (L 47).

*L478 fine-grain should be fine-grained.*
The text has been corrected according to the reviewer's comment (L 412).

---

## Author Comment (AC3) · 12 Feb 2021

We are grateful for very detailed and constructive comments from Referee 3. We enclose a full account of our responses (normal font) to the comments and suggestions of the reviewers (*italics*), including references to all changes in the manuscript.

*General comments: This is an interesting manuscript that used multiple lines of evidence, including element, isotope, and water mass balances to inform their interpretation. It will be a valuable contribution to the scientific literature for its insights into both silicon biogeochemistry in these unique high arctic lake environments and the importance of groundwater as a source of dissolved silicon to lakes. I appreciated the scenario analyses that were undertaken for some of the calculations considering the*

*limitations to the dataset.*

*I believe that the manuscript is of sufficiently high quality in terms of its methods and interpretation of its results and is a novel contribution to Biogeosciences. I have some minor comments for the authors to consider before it is ready for publication in Biogeosciences.*

*Specific comments:*
*Should appendices A, B and C be placed in the supplementary materials rather than in the main manuscript text file?*
Based on BG definition of Appendices (see below), we think that the content should remain part of Appendices rather than Supplementary Materials, as the scenarios and the model calculations are important and necessary for understanding the manuscript.

"**Appendices:** all material required to understand the essential aspects of the paper such as experimental methods, data, and interpretation should preferably be included in the main text. Additional figures, tables, as well as technical and theoretical developments which are not critical to support the conclusion of the paper, but which provide extra detail and/or support useful for experts in the field and whose inclusion in the main text would disrupt the flow of descriptions or demonstrations may be presented as appendices."

"**Supplementary material** is reserved for items that cannot reasonably be included in the main text or as appendices. These may include short videos, very large images, maps, CIF files, as well as short computer codes such as matlab or python script. In no case can supplementary material contain scientific interpretations or findings that would go beyond the contents of the manuscript. In general, supplementary material that can be hosted in alternative sites such as FAIR-aligned data repositories should be placed there. These include data sets, movies, animations, or computer programme codes, for which a persistent identifier, ideally a DOI, should be mentioned in the "data availability" section of the manuscript. Normal size figures, tables, as well as technical or theoretical developments that do not need to be included in the main text should be included as appendices."

*I recommend using the term "isotope ratio" instead of "isotopic signature" throughout the manuscript. I refer to Waterbirds, 35(2):324-331 (2012). https://doi.org/10.1675/063.035.0213 for isotope ratio reporting guidelines, which summarizes the recommendations of the Commission on Isotopic Abundances and Atomic Weights (CIAAW) of the International Union of Pure and Applied Chemistry (IUPAC).*

All instances of "isotope signature" were replaced by "isotope ratio".

*Lines 3-4: Can you be more specific instead of just saying "factors and mechanisms". I realize that this is just the abstract.*
This sentence has been removed from the abstract, and new wording is indicated in the next point.

*Line 4: I somewhat disagree with the statement "While BSi formation and preservation is expected to occur in silica rich environments with high dissolved silicon (DSi) concentrations such as volcanic and hydrothermal inputs, the factors and mechanisms explaining high DSi and BSi concentrations in lakes remain unclear." What qualifies as a high DSi concentration? DSi concentrations are relatively high (in the 100 uM, or 6 mg SiO2 l-1, range) in most groundwater, rivers, and lakes that drain catchments with siliceous bedrock and soils, but also in most catchments overall. BSi formation and preservation also happens in these environments. I understand that this is just the abstract and you qualify these statements more in the introduction. Obviously, as you point out and discuss, the BSi concentrations in the lake's sediments are higher than many other lakes because of the low sedimentation rate, which is unique to high arctic lakes. But I don't think that you can say that the high DSi and BSi concentrations you observed in the lake water column are abnormally high compared to a temperate or tropical freshwater lake.*

*Even in your short summary on Biogeosciences, you say "The processes responsible for large accumulation of siliceous shells of a single-cell algae – diatoms – in lake deposits are poorly understood." I would somewhat disagree with this statement as well. BSi preservation occurs when BSi production and deposition is greater than its dissolution. To me, it seems that the more specific description of the research gap is that "the Si budget of this the lake had not been fully characterized before to establish the drivers of BSi accumulation in this environment, which is a more specific description of what your research objectives were rather than just saying "that the processes responsible for BSi accumulation in sediments are poorly understood".*

Former Lines 4-6 were changed as follows: We explored the factors responsible for the high BSi concentration in sediments of a small, high-latitude subarctic lake (Lake 850). The Si budget of this lake had not been fully characterized before to establish the drivers of BSi accumulation in this environment. (Lines 2-4)

*Lines 30-31 What physical processes are you referring to that remove DSi from solution? Can you be more specific?*
The text has been changed according to the reviewer's comment. We have removed physical and replaced it by chemical processes. The text reads now: "where it can be removed by biological or chemical processes, such as secondary clay mineral formation or amorphous silica precipitation (Jenny, 1941)." (Line 30)

*Line 117: Regarding "and 1000 is unit conversion.": Please specify what the units of this unit conversion factor are.*
We have specified the units of this unit conversion factor: "...of the lake $cm^2$ and 1000 is the unit conversion from g to mg." (Line 124)

*Line 119: When you say "Assuming steady-state ($\Delta$DSi = 0)": Can you elaborate more and be more specific for readers not familiar with steady state conditions what you are talking about. Something to the effect of "Assuming that the DSi fluxes in the lake are at steady state (i.e., the sum of the input fluxes is equal to the sum of the output fluxes and therefore that $\Delta$DSi = 0)"*
We have included a definition of steady-state conditions: "Assuming that the lake is in steady-state, which means that the sum of input DSi fluxes equals to the sum of output Si fluxes, thus $\Delta_{DSi} = 0$, DSi concentration in groundwater was ..." (Lines 125-126)

*Lines 181-186: What method did you use to measure the DSi content in your alkaline extractions? I assume that it was by the automated molybdate-blue method that was used to measure the DSi content in you water samples, but please add this detail to clarify for readers. Please also give a bit more detail about the extraction method and its requirement for a mineral correction factor. You could even do this by saying "Si-*

*containing minerals" on line 184 instead of just "minerals"*
We have modified the text to provide information about the method used for alkaline extraction: Now the text reads as: "The extracted DSi was measured using the automated molybdate-blue method (Strickland, 1972) with a Smartchem 200, AMS System™ discrete analyzer at Lund University with an instrumental error of $\pm 3.7\%$. As there were no changes in the amount of total Si extracted during the time course of dissolution (n=3, slope $\sim 0$), the mean BSi concentration from all the values was used to estimate BSi concentration with no Si-containing minerals correction applied (Conley, 1998)." (Lines 191-195)

*Lines 403-408: Could these lines go in the Model uncertainties discussion section given that you have a section dedicated to discussing these uncertainties?*
We moved this paragraph to Model uncertainties section according to the reviewer's comment (Lines 461-467)

*Lines 471-480: I think this discussion, which is the conclusions, belongs in its own discussion section about comparing your lake's environmental controls on BSi accumulation with that of other lakes and environments where high BSi accumulation has been observed, and the implications of this finding (regarding the fact that groundwater is likely a significant source of DSi to lakes and should be considered in lake Si budgets). Then, one sentence in your conclusions can summarize this discussion section.*
We followed the reviewer's suggestion and moved this paragraph into the new Discussion section "Environmental controls on BSi accumulation" (Lines 405-412).

Then a brief summary of this paragraph was placed in the conclusions section: Lakes on silica-rich bedrock, with low allochthonous input, low sedimentation rates, low-relief watershed geomorphology and groundwater input have a high potential to accumulate BSi. (Lines 495-496)

*Lines 474-476: Reconsider your discussion about high autochthonous carbon production in arctic lakes here. Autochthonous carbon production rates (that is, carbon fixation*

*rates) are not independent of diatom DSi uptake rates and BSi production and burial rates.*

*A related thought, which might be outside of the scope of this work, but I thought I'd share nonetheless: Have you thought about the implications of your work for our understanding of carbon burial in arctic lakes. Specifically, I am thinking – could the high diatom productivity, driven by high DSi supply via groundwater, drive increased carbon burial in lakes with high groundwater discharge relative to lakes with less groundwater discharge (and therefore less DSi supplied)?*

*Could this be a research hypothesis that you could put forth in this paper for others to answer? In other words, could you suggest that the implications of your work extend beyond understanding Si biogeochemistry and can inform our understanding of carbon fixation and burial rates in arctic lakes? It is especially interesting that you have sediment TOC concentration data, and they appear to be correlated with the sediment BSi concentrations.*

*You could refer to these papers: Wang, B., Liu, CQ., Maberly, S. et al. Coupling of carbon and silicon geochemical cycles in rivers and lakes. Sci Rep 6, 35832 (2016). https://doi.org/10.1038/srep35832 Krause, J.W., Schulz, I.K., Rowe, K.A. et al. Silicic acid limitation drives bloom termination and potential carbon sequestration in an Arctic bloom. Sci Rep 9, 8149 (2019). https://doi.org/10.1038/s41598-019-44587-4*
We are grateful for the very interesting suggestion in this comment. The TOC can be partly connected to BSi production. However, in Lake 850 we have a large production of aquatic mosses, which most likely causes that the TOC not being correlated with BSi. Thus, we do not have enough supportive data to open the discussion on the carbon burial through diatoms. Nevertheless, very interesting comment and possible scope for future studies.

*A general technical comment: Check your pluralization of words and whether you need an article (a/an or the) in front of your nouns. For example, I would recommend that*

*stream discharge does not need to be pluralized. I made note of some of these gram-mar/technical corrections, but not of all of them.*
We did change stream discharges to stream discharge in the entire document and reviewed the pluralization as suggested.

*Line 1: "concentrations occur" instead of "concentration occurs"*
Corrected, Line 1.

*Line 6: "stream discharge" instead of "stream discharges"*
Corrected, now Lines 4-5.

*Lines 8-9: I would recommend re-phrasing to clarify that one fifth of the DSi that would otherwise be exported to the ocean is retained by lakes, something like: "estimated to retain one fifth of the annual DSi terrestrial weathering flux that would otherwise be delivered to the ocean" instead of "estimated to retain one fifth of the annual DSi delivery into the ocean"*
We applied reviewer's suggestion and text was modified (Lines 11-12)

*Line 9: "DSi inputs being 3 times higher" instead of "DSi inputs 3 times higher"*
Corrected, now Line 7.

*Line 17: "dissolved silicic acid, H4SiO4, expressed here as dissolved silicon (DSi), and . . ." instead of "dissolved silicic acid H4SiO4, expresses here as dissolved silicon (DSi), and . . ."*
Corrected, now Line 16.

*Line 23: "One example is high-elevation" instead of "One example is the high-elevation"*
Corrected, now Line 22.

*Line 24: "high BSi concentrations" instead of "large BSi concentrations"*
Corrected, now Line 23.

*Line 30: "removed" instead of "taken up" (given that you refer to both physical and biological processes)*

Corrected, now Line 29.

*Lines 45-46: "Therefore, stable Si isotopes are an" instead of "Therefore, the stable Si isotopes provide an"*
Corrected, now Line 45.

*Line 60: I think you mean "850 m above the tree-limit" instead of "850 m a.s.l., above the tree-limit"*
Text has been changed to "Lake 850 lies above the tree-limit (600 m a.s.l) at 850 m a.s.l." (Lines 60-61)

*Line 184: Change wording: "As there were no changes in the amount of total Si extracted during the time course of dissolution" instead of "As no changes in the amount of total Si extracted during the time course of the dissolution"*
Text was changed to: "The extracted DSi was measured using the automated molybdate-blue method (Strickland, 1972) with a Smartchem 200, AMS System™ discrete analyzer at Lund University with an instrumental error of $\pm 3.7\%$. As there were no changes in the amount of total Si extracted during the time course of dissolution (n=3, slope $\sim 0$), the mean BSi concentration from all the values was used to estimate BSi concentration with no Si-containing minerals correction applied (Conley, 1998)." (Lines 191-195)

*Line 301: "accumulation" instead of "accumualtion"*
Corrected, now Line 312.

*Line 363: "The production consumes 63%" instead of ": "The production consumes from 63%"*
Corrected and numbers updated, now Line 376.

*Line 381: "The significance of groundwater-sourced DSi to the lake's Si cycle" or "The significance of groundwater-sourced DSi to Lake 850's Si cycle" instead of "The significance of groundwater on lake Si cycle"*

Corrected to: "The significance of groundwater-sourced DSi to the lake's Si cycle is..."
(Line 396)

*Lines 410-435: Please specify every time that you mention the modelled groundwater
d30Si values that they are modelled. For example: please add this specification to
lines 416 and 417.*
Corrected and specified in paragraphs on Lines 433-453.

*Figures:*
*You use the shortform HTH core in the caption for Figures 3 and 4 but you don't explain
this shortform in the text. Please change this.*
We exchanged "HTH" for "gravity" core to make the caption clear.

*Can you increase the font size for the axis labels and numbers in Figure 3? And could
the inset figures go in the Supplementary Information? They seem extraneous and are
not thoroughly explained or discussed in the main text. I understand if this is difficult to
do because of the default graphical output of the PLUM package.*
The font size was increased, and insets were removed. The full figure added to supplementary files as Figure S2.

*Can you increase the resolution of Figure 5? It seems to be a lower resolution compared to the other figures, and the x-axis numbers are somewhat cut off at the top.*
Corrected.